# Neuroinflammation in neuronopathic Gaucher disease: Role of microglia and NK cells, biomarkers, and response to substrate reduction therapy

Chandra Sekhar Boddupalli[1†], Shiny Nair[1†], Glenn Belinsky[1], Joseph Gans[2], Erin Teeple[2], Tri-Hung Nguyen[2], Sameet Mehta[3], Lilu Guo[2], Martin L Kramer[2], Jiapeng Ruan[1], Honggge Wang[2], Matthew Davison[2], Dinesh Kumar[2], DJ Vidyadhara[4], Bailin Zhang[2], Katherine Klinger[2], Pramod K Mistry[1,5]*

[1]Department of Internal Medicine, Yale School of Medicine, New Haven, United States; [2]Translational Sciences, Sanofi, Framingham, United States; [3]Yale Center for Genome Analysis, Yale School of Medicine, New Haven, United States; [4]Department of Neuroscience, Yale School of Medicine, New Haven, United States; [5]Department of Molecular & Cellular Physiology, Yale School of Medicine, New Haven, United States

*For correspondence:
pramod.mistry@yale.edu

†These authors contributed equally to this work

## Abstract:

**Background:** Neuronopathic Gaucher disease (nGD) is a rare neurodegenerative disorder caused by biallelic mutations in *GBA* and buildup of glycosphingolipids in lysosomes. Neuronal injury and cell death are prominent pathological features; however, the role of *GBA* in individual cell types and involvement of microglia, blood-derived macrophages, and immune infiltrates in nGD pathophysiology remains enigmatic.

**Methods:** Here, using single-cell resolution of mouse nGD brains, lipidomics, and newly generated biomarkers, we found induction of neuroinflammation pathways involving microglia, NK cells, astrocytes, and neurons.

**Results:** Targeted rescue of *Gba* in microglia and neurons, respectively, in *Gba*-deficient, nGD mice reversed the buildup of glucosylceramide (GlcCer) and glucosylsphingosine (GlcSph), concomitant with amelioration of neuroinflammation, reduced serum neurofilament light chain (Nf-L), and improved survival. Serum GlcSph concentration was correlated with serum Nf-L and ApoE in nGD mouse models as well as in GD patients. *Gba* rescue in microglia/macrophage compartment prolonged survival, which was further enhanced upon treatment with brain-permeant inhibitor of glucosylceramide synthase, effects mediated via improved glycosphingolipid homeostasis, and reversal of neuroinflammation involving activation of microglia, brain macrophages, and NK cells.

**Conclusions:** Together, our study delineates individual cellular effects of *Gba* deficiency in nGD brains, highlighting the central role of neuroinflammation driven by microglia activation. Brain-permeant small-molecule inhibitor of glucosylceramide synthase reduced the accumulation of bioactive glycosphingolipids, concomitant with amelioration of neuroinflammation involving microglia, NK cells, astrocytes, and neurons. Our findings advance nGD disease biology whilst identifying compelling biomarkers of nGD to improve patient management, enrich clinical trials, and illuminate therapeutic targets.

**Funding:** Research grant from Sanofi; other support includes R01NS110354.

## Editor's evaluation

The pathophysiology of neuropathic Gaucher disease (nGD) is far from clear. This study not only provides mechanistic insights into the processes underpinning microglial activation and neuroinflammation but also implicates glycosylceramide in the pathogenesis of nGD through the use of a novel inhibitor. Overall, the landmark study substantially improves our understanding of nGD with significant therapeutic ramifications.

## Introduction

In Gaucher disease (GD), biallelic mutations in GBA underlie defective acid β-glucosidase (glucocerebrosidase, GCase) and buildup of the primary substrate, glucosylceramide (GluCer), and its inflammatory metabolite glucosylsphingosine (GlcSph) in the lysosomes (*Grabowski et al., 2021*). GD is classified into three broad phenotypes based on the absence (type 1 GD, GD1) or onset of early-onset neurodegeneration in childhood (fulminant GD type 2, GD2, and slower chronic neurodegeneration, GD3). Adults with GD1 have a markedly increased risk of Parkinson's disease and Lewy body dementia (PD/LBD) (*Bultron et al., 2010*; *Grabowski et al., 2021*). Moreover, heterozygous carriers of GBA mutations are at an increased risk of PD/LBD (*Aharon-Peretz et al., 2004*; *Sidransky and Lopez, 2012*; *Sidransky et al., 2009*). Notably, low GCase is reported in the brains of sporadic PD patients who do not harbor GBA mutations (*Gegg et al., 2022*). Together, GBA variants and GCase deficiency are important determinants of a wide spectrum of neurodegenerative diseases.

Glucosylceramide and glucosylsphingosine display potent inflammatory and immunogenic activities (*Nagata et al., 2017*; *Nair et al., 2015*; *Nair et al., 2016*; *Pandey et al., 2017*). Treatment of non-neuronopathic GD1 involves enzyme replacement therapy (ERT) targeted to the macrophages via mannose receptors and substrate reduction therapies (SRTs) using inhibitors of glucosylceramide synthase (GCS) (*Cox et al., 2000*; *Mistry et al., 2017a*; *Mistry et al., 2017b*; *Platt et al., 2018*). However, currently, there are no effective therapies for the devastating neurodegenerative sequela of GBA mutations. In GD1, SRT is predicated on the concept that the rate of synthesis of GluCer is reduced to match residual enzyme activity due to GBA mutations (*Platt et al., 2018*). However, in neurodegenerative GD2 and GD3, residual glucocerebrosidase activity is profoundly depressed due to severe GBA mutations. A randomized-controlled trial of brain-penetrant SRT, N-butyldeoxynojirimycin. in patients' GD3 was unsuccessful in ameliorating neurological manifestations. However, N-butyldeoxynojirimycin is a relatively weak inhibitor of GCS with significant off-target effects (*Schiffmann et al., 2008*). Using a more specific, brain-penetrant GCS inhibitor in a chemically induced model of nGD, some disease pathways were ameliorated on bulk RNA analysis, but had no effect on inflammatory pathways (*Blumenreich et al., 2021*).

Microglia are specialized, self-renewing, CNS-resident macrophages that represent the dominant immune cells involved in maintaining CNS homeostasis. Emerging data from single-cell analysis and genome-wide association studies of several neurodegenerative diseases have revealed a central role of microglia in neurodegeneration (*Chen and Colonna, 2021a*). Although several mechanisms have been proposed to explain neurodegeneration associated with GBA mutations, most studies in GD have focused on GBA deficiency in neurons, while contribution of other cell types in driving the disease pathology has been considered minor (*Cho et al., 2019*; *Vitner et al., 2015*). Immunohistochemical evidence of microglial alteration have been reported in neuronopathic GD, but the effect of GBA deficiency in microglia per se is not understood. Further studies are warranted to investigate the immune landscape of nGD to clearly delineate therapeutic targets and generate biomarkers.

Hitherto, studies aimed at understanding the mechanisms of neuroinflammation in neurodegeneration due to GBA deficiency have solely relied on bulk cell population analysis, which have hindered the delineation of heterogeneity and complexity of the immune milieu within the brain as well as the role of GBA in individual cell types of the brain. Here, we applied an integrated approach based on novel mouse models, lipidomic analyses, scRNA-seq of immune cells, and brain snRNA-seq to decipher temporospatial components of neuroinflammation associated with *GBA*, deficiency including immune cell subsets, activated pathways, as well as probe therapeutic targets and discover novel biomarkers. Importantly, we used both early-onset neuronopathic GD mice *Gba*[lsl/lsl] (loxP-stop-loxP) mice, with a germline deletion of *Gba* (henceforth referred to as nGD mice) which phenocopies human GD2, fulminant neuronopathic GD (*Enquist et al., 2007*) as well as a new mouse model of microglia-specific *Gba*

deletion in *Gba^{fl/fl}* mice that mimics the late-onset progression seen in some patients with GD1. To further understand the function of *Gba* in microglial and neuronal homeostasis and disease progression, we developed additional new nGD mouse models with microglia and neuron-specific rescue of *Gba*.

We evaluated brain-penetrant inhibitor of GCS as a therapeutic strategy in both early- and late-onset nGD models to assess its impact on reducing cellular glycosphingolipids, individual components of neuroinflammation and neurodegeneration while elucidating temporospatial cellular events. Our findings define a previously unreported role of glycosphingolipid-laden microglia and macrophages along with NK cells and astrocytes in neurodegeneration associated with *Gba* deficiency while also revealing novel biomarkers and therapeutic targets.

# Methods

**Key resources table**

| Reagent type (species) or resource | Designation | Source or reference | Identifiers | Additional information |
|---|---|---|---|---|
| Antibody | CD11b | BD Biosciences | 583,553 | (1:1000) |
| Antibody | CD11b | BD Biosciences | 552,850 | (1:1000) |
| Antibody | ACSA-2 | Miltenyi Biotech | 130-123-284 | (1:100) |
| Antibody | CD45 | Biolegend | 103,133 | (1:1000) |
| Antibody | NK1.1 | BD Biosciences | 562,864 | (1:250) |
| Antibody | CCR2 | R and D | FAB5538F | (1:100) |
| Antibody | CD3 | Biolegend | 100,241 | (1:250) |
| Antibody | NK1.1 | Biolegend | 108,753 | (1:250) |
| Antibody | 1 A/1-E | Biolegend | 107,639 | (1:1000) |
| Antibody | CD4 | BD Biosciences | 563,790 | (1:500) |
| Antibody | CD4 | BD Biosciences | 558,107 | (1:1000) |
| Antibody | Gr-1 | Biolegend | 108,440 | (1:1000) |
| Antibody | Ly-6C | Invitrogen | 47-5932-82 | (1:1000) |
| Antibody | CD64 | Biolegend | 139,306 | (1:200) |
| Antibody | B220 | eBioscience | 47-0452-80 | (1:200) |
| Antibody | CD8a | eBioscience | 56-0081-82 | (1:1000) |
| Antibody | IL-1Beta | Invitrogen | 17-7114-80 | (1:1000) |
| Antibody | IFNγ | Biolegend | 505,808 | (1:250) |
| Antibody | IFNγ | BD Biosciences | 562,020 | (1:250) |
| Other | Brdu Flow kit | BD Pharmingen | 51-9000019AK | |
| Other | 5-Bromo-2'-deoxyuridine | Cayam Chemical Company | 15,580 | 180 µg/ml (injected) and 800 µg/ml (in water) |
| Other | Cell Rox Deep Red kit | Invitrogen | C10422 | |

## Patients

Stored serum samples of GD patients were analyzed. All patients had diagnosis of GD based on <10% of normal leucocyte acid-β glucosidase activity. The study was approved by the Human Investigations Committee of Yale School of Medicine. 51 patients had type 1 GD with at least one N370S (pArg409Ser) mutation in GBA gene, mean age 63.9 years (range 40–93 years). Patients were stratified into young and older patients with mean age 11.25 years vs. 55.8 years. Five patients had GD type 3 (homozygous for L444P mutation, pLeu483Leu). The mean age of 28 healthy controls was 38 years.

## Mice

Mice were housed in the animal facility of Yale University in New Haven. All animal experiments were conducted in compliance with institutional regulations under authorized protocol (2016-10872) approved by the Institutional Animal Care and Use Committee. $Gba^{lsl/lsl}$ mice or $Krt14^{lnl/lnl}$ were generated as described previously (**Enquist et al., 2007**). We refer to $Gba^{lsl/lsl}$ as nGD mice throughout the text. We used $Krt14^{Cre}$ (strain #: 018964), $Cx3cr1$ Cre (strain #: 025524), and $Nestin$ Cre (strain #: 003771) obtained from Jackson Laboratories. For breeding purpose, we used $Gba^{lnl/wt}$ mice (gift from Sanofi Genzyme) that were then crossed with $Krt14^{Cre/Cre}$ (Jackson Laboratories) to obtain $Gba^{lsl/wt}$ $Krt14^{cre/cre}$. These mice were used as parents to obtain nGD, $Gba^{lsl/wt}$, and $Gba^{wt/wt}$ pups. nGD $Cx3cr1^{Cre/+}$ and nGD $Nes^{Cre/+}$ were obtained by breeding these parents on to $Cx3cr1$ Cre and $Nestin$ Cre. $Gba^{fl/fl}Cx3cr1^{cre}$ mice were generated by crossing $Gba^{fl/fl}$ (**Mistry et al., 2010**) with $Cx3cr1$ Cre mice obtained from Jackson Laboratories. Mice of both sexes were used for the study. From postnatal day 8, C57BL/6J mice (Jackson Laboratories, USA) were injected intraperitoneally (i.p.) daily with 25 mg conduritol β-epoxide (CBE) (Calbiochem Millipore, Darmstadt, Germany) per kilogram body weight, or with phosphate-buffered saline (PBS) (**Vardi et al., 2020**). Newborn mice were gavaged daily with 5 mg/kg of GZ 161 once a day starting at postnatal day 4 (**Cabrera-Salazar et al., 2012**). To determine whether further benefits could be achieved by prenatal exposure to GZ 161, a subset of mice also received GZ 161 in food using a formulation calculated to provide 20 mg/kg/day.

### Brain tissue harvesting and cell preparation

The complete brain was cut into small pieces and incubated with digestion buffer (RPMI supplemented with 2% FBS, 2 mM HEPES, 0.4 mg/ml Collagenase D, and 2 mg/ml DNase) for 30 min at 37°C under shaking. To stop enzymatic digestion, EDTA 5 mM was used and the sample was homogenized with a syringe. This was then followed by gradient centrifugation, and cell separation was achieved via Percoll gradients (GE Healthcare Life Sciences) of various densities (**Lee and Tansey, 2013**). Myelin was then removed by vacuum suction and cells were isolated from the interphase. The interphase was diluted with 10 ml of HBSS and centrifuged at 500 × $g$ for 7 min to obtain cell pellet that was used for cellular analysis.

### Flow cytometry

Flurochrome-labeled antibodies used for the flow cytometry are listed in the Key resources table. In brief, surface staining was performed ex vivo using antibodies to CD11b, CD45, Gr-1, CD3, CD8, CD4, CD103, CD69, and NK1.1 (BD Biosciences); ACSA-2 antibody (Miltenyi Biotech). For staining of the intracellular antigens, GZM-A and Pro IL-1β (BD Biosciences) cells were stimulated with cell stimulation cocktail (eBioscience) for 4 hr. After surface staining with antibodies, cells were fixed and permeabilized using BD cytofix and Cytoperm and antibodies recognizing GZM-A and Pro IL-1β.

## Sorting of cells by flow cytometry

Isolated total brain cells were resuspended at 1 million cells/ml of PBS and Live/Dead Fixable Violet Dead Cell Stain (Thermo Fisher) for 10 min to exclude nonviable cells. Cells were washed once in excess PBS, and then, cells were suspended in ice-cold FACS buffer (10% FBS in PBS with 1% HEPES + 0.5% EDTA) and stained with anti-mouse CD45 and anti-mouse CD11b antibody (BD Pharmingen) for 30 min at 4°C. All cells were washed twice with FACS buffer and sorted into polypropylene tubes with 500 µl of ice-cold FACS buffer. All samples were acquired on the BD FACS Aria.

### Single-cell RNA sequencing (scRNA-seq)

All cells were prepared through 10X Genomics V3 3′ Gene Expression kit and sequenced on NovaSeq flow cells to achieve high read depth. We used the preprocessed digital gene expression matrices as obtained from cell ranger (as run by the YCGA sequencing core facility). These matrices were processed using Seurat (**Satija et al., 2015**). The clusters were determined using Louvain algorithm for community detection (**Vincent et al., 2008**). Differential gene expression between the clusters was carried out using the MAST method (**Finak et al., 2015**). The visualizations and analysis were carried out in R (R Core Team). The data were visualized as t-distributed stochastic neighbor embedding (t-SNE) (**Hinton, 2002**). From 14-day pups, we sequenced a total of 4895 CD45+ cells. For GZ161

treatment experiments, we sequenced a total of 40,763 CD45+ cells that consists of four groups of mice with three mice per each group.

## Subsetting of microglia cells

Based on the clustering of CD45⁺ cells, clusters 1, 16, and 26 were kept for final analysis. The clusters were selected based on the microglial gene expression. The clusters comprised 4230 cells. Scoring used z-scores of homeostatic microglia genes, DAM genes from *Wang et al., 2020*.

## Single-nuclear RNA sequencing (snRNA-seq)

### Nuclei isolation

Brain tissue samples were stored at –80°C. For tissue lysis and washing of nuclei, sample sections were added to 1 ml lysis buffer (Nuclei PURE Lysis Buffer, Sigma) and thawed on ice. Samples were then Dounce homogenized with PestleAx20 and PestleBx20 before transfer to a new tube, with the addition of additional lysis buffer. Following incubation on ice for 15 min, samples were then filtered using a 30 μM MACS strainer (MACS strainer, Fisher Scientific), centrifuged at 500 × *g* for 5 min at 4°C using a swinging bucket rotor (Sorvall Legend RT, Thermo Fisher), and then pellets were washed with an additional 1 ml cold lysis buffer and incubated on ice for an additional 5 min. Lysates were combined with 1.8 ml of a 1.8 M sucrose solution (Nuclei PURE Sucrose Buffer, Sigma) containing 1 mM DTT and 0.2 U RNAse inhibitor and mixed by inversion. Samples were then layered on top of a 1.8 M sucrose layer to form a gradient and centrifuged at 30,000 × *g* for 45 min at 4°C using a swinging bucket rotor. The supernatant was removed, and nuclei pellets were resuspended in 1 ml of a 1× PBS wash buffer containing 1% BSA and 0.2 U RNAse inhibitor. Samples were centrifuged again at 500 × *g* for 5 min at 4°C and the supernatant removed. Nuclei were resuspended in 0.5 ml wash buffer and counted using Countess (Life Technologies) prior to 10X Genomics protocol. For samples that were enriched using flow cytometry, nuclei were stained with 20 μg/ml DAPI for 30 min on ice with occasional mixing. Nuclei were centrifuged at 500 × *g* for 5 min at 4°C. The supernatant was removed, and pellet resuspended in 800 μl of a 1× PBS wash buffer containing 1% BSA, 1 mM EDTA, and 0.2 U RNAse inhibitor. Nuclei were sorted using the BD Influx by first gating on forward/side scatter then on DAPI-positive nuclei. The collected nuclei were centrifuged, then resuspended in ~200 μl 1× PBS wash buffer containing 1% BSA and 0.2 U RNAse inhibitor and counted using Countess (Life Technologies) prior to 10X Genomics protocol.

### Library preparation and NovaSeq sequencing

Libraries were prepared according to 10X Genomics protocol using the Chromium Next GEM Single Cell 3′ Reagents Kit V3.1 (Dual Index) for encapsulation, mRNA capture, cDNA synthesis/amplification, and library construction. Final libraries were quantified using the DNA High Sensitivity Kit (Agilent Bioanalyzer 2100) and Qubit 2.0 (Life Technologies). Libraries were diluted to 1.5 nM, then pooled prior to sequencing on the Illumina NovaSeq6000. UMI count matrices were generated with Cell Ranger V3.0.2.

### Data preprocessing

Summary information for final UMI count matrices for nuclei by individual sample and sequencing data is presented in (*Figure 5—figure supplement 1A, C*). Count matrices together with nucleus barcodes and gene labels were loaded with R version 3.6.1/RStudio for sample integration and unsupervised clustering using Seurat Package version 3.1. For quality control (QC), nuclei were filtered following standard protocols based on examination of violin plots. Cutoffs were used, and the filtered matrices were then individually log-normalized by sample according to standard Seurat workflows. After quality filtering, total nuclei were included in the final data analysis.

### Broad cell-type annotation

Sample integration was performed in Seurat using the Find Integration Anchors and Integrate Data functions for variable features. Following integration and scaling according to Seurat package workflows, a range of clustering resolution values were trialed prior to broad cell-type annotation; UMAP, t-SNE plots for broad types annotation included. Cluster-level expression of major cell type

markers was examined and used to annotate cells contained within each cluster. Clustering resolution achieved separation and consistency of cell-type marker expression.

## Differential gene expression and pathway enrichment analysis

Genes differentially expressed between conditions within clusters may reveal pathway alterations specific to the clustered cell type. To profile cluster-level pathway enrichment patterns, cluster-level differentially expressed genes were identified using the Seurat Find Markers function with the MAST package. Markers used for downstream functional analysis were those with adjusted p-value <0.05. Functional ontology analysis by cluster was performed for each cluster marker set using ingenuity pathway analysis (IPA) (enrichment-adjusted p-value < 0.05).

## Total RNA-seq

mRNA sequence data were uploaded to a high-performance computing system with PartekFlow software (v7.0, Partek, St. Louis, MO), adapter trimmed and remapped to mouse genome, mm10 using STAR v2.5.3a aligner with default setting (Phred:20) for read mapping. Statistical analysis was carried out using false discovery rate (FDR) correction through the Benjamini–Hochberg method.

## BrdU retention assay

The assay was performed as described earlier (*Takizawa et al., 2011*); briefly mice were i.p. injected with 180 µg BrdU (sigma) and were fed water containing 800 µg/ml BrdU and 4% glucose for 12 days. Mice were then sacrificed, organs (brain and spleen) were removed, and cells were isolated as described earlier. BrdU staining was performed using BrdU labeling kit (BD). Cells from normal water-fed mice were used as staining controls.

## Mouse serum Nf-L assay

Quanterix Nf-L assays were performed in triplicate according to the manufacturer's protocols using the Nf-Light Advantage kit on a single-molecule array (SIMOA) HD-X instrument (Quanterix, Lexington, MA).

## Lipidomics

Separation of glucosylceramides and galactosyl ceramides was performed by SFC-MS/MS analyses at the MUSC Lipidomics Shared Resource. The equipment consisted of a Waters UPC 2 system coupled to a Thermo Scientific Quantum Access Max triple quadrupole mass spectrometer equipped with an ESI probe operating in the multiple reaction monitoring-positive ion mode tuned and optimized for the Waters UPC 2 system.

## Brain tissue slide preparation and MALDI-MS imaging

Brain tissues were dissected, divided sagittally into halves and immediately placed in Cryomolds (Tissue-Tek) containing 10% gelatin-water (porcine skin, Sigma-Aldrich, #G1890), placed at 37°C on the heating block, and transferred to the cryomold into dry-ice bath. Cryo-sectioning was performed at a chamber temperature of –20°C with 12 µm thickness. Sections were thaw-mounted onto the ITO-coated side with barcoding of MALDI IntelliSlides (Bruker, Cat# 1868957). Slides were stored at –80°C until time for imaging. An optical image for tissue sections was obtained using TissueScout scanner (Bruker). Once the optical image was obtained, it was transferred to the HTX Sublimator (HTX Technologies, Chapel Hill, NC) for matrix deposition. DHB (2,5-dibydroxybenzoic acid) was applied on the tissue sections by sublimation, which was performed according to the HTX Sublimator setting (2 ml of 40 mg/mL DHB in acetone transfer solvent, 60°C preheat temperature, 160°C final temperature, 200 s). MALDI-MS imaging was performed in positive ion mode over a mass range of m/z 300–1300 on a Bruker timsTOF fleX mass spectrometer equipped with a 10 kHz SmartBeam 3D Nd:YAG (355 mm) laser. Imaging was performed using a laser raster size of 20 µm custom setting, 20 µm scan range, with trapped ion mobility mode On (tims'ON') for cross-sectional collision values with the following parameters: $1/K_o$ start at 0.90 V.s/cm$^2$ and end 1.70 V.s/cm$^2$, ramp time at 150 ms, accumulation time 40.0 ms, duty cycle 26.67%, ramp rate 6.50 Hz. Spectra were accumulated from 400 laser shots with the laser power percentage adjusted using the highest peak (m/z 760 positive ion mode) intensity between $10^4$ and $10^5$, and the method was saved for all subsequent acquisitions. For each image

acquisition, the instruments were calibrated using ESI-L Low concentration tuning mix (Agilent Technologies, Cat #G1969-85000). Data were analyzed using SCiLs Lab MALDI imaging software package, version 2021b (Bruker), with data normalized to the TIC. Targeted analysis for hexceramides, sphingomyelins, and phosphatidylcholines was analyzed for differential changes between animals.

## Lipid extraction of mouse serum

To quantify GlcSph, 7 µl of mouse serum was aliquoted into a labeled 1.5 ml Eppendorf tube followed by 100 µl of internal standard solution (12 ng/ml dimethyl-psychosine, 80% methanol, 20% acetonitrile with 10 mM ammonium acetate, and 1% formic acid). The samples were vortexed for 5 min and sonicated for 10 min. The tubes were centrifuged at 13,000 × $g$ for 5 min at room temperature (RT). The supernatant from each tube was transferred into a pre-labeled total recovery MS vial. Calibration curves for GlcSph were prepared in a pooled control serum, and concentrations of lysoGL1 were from 0.1 to 1000 ng/ml.

## Mass spectrometry analysis of GlcSph

Mouse serum was first tested on a Hilic- method to separate psychosine (steroisomer of GlcSph) and GlcSph and confirm that psychosine does not interfere with the quantitation of mouse serum glucosylsphingosine levels (Wei-Lien Chuang et al.). The supernatant was injected (5 µl) into an LC/MS/MS system comprised of an Acquity UPLC (Waters, Milford, MA) and Sciex Triple Quad 5000 mass spectrometer (Sciex, Toronto, Canada). The chromatographic separation was achieved with a Waters Acquity BEH C18 column (2.1 * 150 mm, 1.7 µm) using mobile phases: (A) water with 0.1% formic acid and (B) 85:15 MeOH:ACN with 0.1% formic acid. The column was maintained at 60°C. GlcSph was eluted with the following gradient: from 50% B to 99% B over 2 min, then the mobile phase composition was hold constant for 1 min followed by a rapid return (0.1 min) to 50% B maintained for 0.5 min. All experiments were carried out at a flow rate of 0.5 ml/min. Data were analyzed in Analyst (AB Sciex, Toronto, Canada).

## Behavioral studies
### Balance beam test

Evaluating fine motor coordination using balance beam (1 mt) with 12 mm width resting 50 cm above the top of the pole. The time to cross each beam was recorded and compared between mouse strains.

## Open field

Mouse locomotor behavior was assessed using the open field (*Crusio, 2001*). Mice were individually placed on the 28 × 28 cm plate surrounded by plastic walls in a well-lit room and allowed to freely explore for 10 min. Spatial statistic, total distance travel, area measure, and time spent in the center of the square were quantified. The movements were recorded with a video tracking system and stored on a computer.

## BODIPY staining

Isolated total mice brain cells from nGD and age-matched control mice were prepared (as described in the section 'Brain tissue harvesting and cell preparation') and stained with CD45, CD11b, and ACSA-2, then incubated in PBS with BODIPY 493/503 (1:1000 from a 1 mg/ml stock solution in DMSO; Thermo Fisher) for 10 min at RT, washed twice in PBS, and BODIPY intensity was analyzed on LSRII instrument (BD Biosciences).

## ROS assay

To assess ROS generation in astrocytes and microglia from nGD mice, isolated total mice brain cells were stained with CD45, CD11b, and ACSA-2 followed by incubation in FACS buffer with CellROX Deep Red (1:500; Invitrogen) for 30 min at 37°C, washed twice in FACS buffer, and CellROX Deep Red Intensity was analyzed on LSRII instrument (BD Biosciences).

## ApoE measurement

ApoE levels in sera of GD patients were measured by ELISA using Abnova Cat# KA1031 following the manufacturer's instructions. A dilution of 1:400 sera was used.

## Quantification and statistical analysis

Data were routinely presented as mean ± SEM. Statistical significance was determined using *t*-test using Bonferroni–Dunn correction for multiple comparisons. Differences between groups were analyzed using Student's *t*-test using GraphPad Prism 8.0. For the Kaplan–Meier analysis of survival, the log-rank (Mantel–Cox) test was performed.

## Study approval

All animal experiments were conducted in compliance with institutional regulations under authorized protocol (2016-10872) approved by the Institutional Animal Care and Use Committee. The study was approved by the Human Investigations Committee of Yale School of Medicine, and written informed consent was received prior to participation.

| Reagents | Supplier name | Cat# | Clone |
|---|---|---|---|
| CD11b | BD Biosciences | 583553 | M1/70 |
| CD11b | BD Biosciences | 552850 | M1/70 |
| ACSA-2 | Miltenyi Biotech | 130-123-284 | IH3-18A3 |
| CD45 | BioLegend | 103133 | 30-F11 |
| NK1.1 | BD Biosciences | 562864 | PK136 |
| CCR2 | R&D | FAB5538F | |
| CD3 | BioLegend | 100241 | 17A2 |
| NK1.1 | BioLegend | 108753 | PK136 |
| 1A/1-E | BioLegend | 107639 | M%/114.152 |
| CD4 | BD Biosciences | 563790 | GK1.5 |
| CD4 | BD Biosciences | 558107 | RM4-5 |
| Gr-1 | BioLegend | 108440 | RB6-8c5 |
| Ly-6C | Invitrogen | 47-5932-82 | HK1.4 |
| CD64 | BioLegend | 139306 | X54-5 |
| B220 | eBioscience | 47-0452-80 | RA3-6B2 |
| CD8a | eBioscience | 56-0081-82 | 53-6.7 |
| Brdu Flow kit | BD Pharmingen | 51-9000019AK | |
| 5-Bromo-2'-deoxyuridine | Cayman Chemical Company | 15580 | |
| Cell Rox Deep Red | Invitrogen | C10422 | |
| IL-1Beta | Invitrogen | 17-7114-80 | NTTEN3 |

## Results

### Glycosphingolipid-laden microglial activation and immune cell infiltration drive GD-associated neurodegeneration

We performed temporospatial analysis of brain in nGD mice (*Gba*$^{lsl/lsl}$ mice with germline deletion of *Gba*, rescued from lethal skin phenotype using *Krt14*$^{Cre}$) compared to control mice (*Gba*$^{lsl/wt}$ and *Gba*$^{wt/wt}$) on days 2, 4, 8, 10, and 14. nGD mice looked clinically well during the first week and subsequently developed progressive ataxia, weight loss, and hind limb paralysis, by day 14, as described previously (*Enquist et al., 2007*). Microglia and macrophage subsets in brain were defined by combinatorial expression of CD11b,

CD45, T cell immunoglobulin and mucin domain containing 4 (TIMD4), and the chemokine receptor C-C motif chemokine receptor 2 (CCR2). Brain microglia have self-renewal capacity with minimal monocyte input in contrast to CCR2+ macrophages, which have limited self-renewal capacity and are constantly replaced by blood monocytes (*Dick et al., 2022*). Infiltration of CCR2+ macrophages (MFs) defined as CD11b^hi CD45+CCR2+ CD64+ TIMD4- population (*Figure 1—figure supplement 1A*) was noted in the nGD brain from an early time point day 2, which showed a steady increase until day 14 when the mice reached the humane end point (*Figure 1A*). Concurrently, there was attrition of homeostatic microglia and incremental infiltration of diverse immune cells into nGD brains, which coincided with mice displaying clear signs of neurodegeneration (*Figure 1A and B*, *Figure 1—figure supplement 1A*). The predominant immune cells in healthy mouse brains were homeostatic microglia, as expected, whereas in nGD brains, the repertoire of immune cells comprised diverse lymphoid (T cells, NK cells, ILC-2, pDC), and myeloid compartment (cDC, monocytes, and/or macrophages) (*Figure 1—figure supplement 1B*). To investigate the metabolic consequences of *Gba* deletion in microglia and infiltrating immune cells, we performed HPLC/MS/MS on flow-sorted microglia and infiltrating immune cells. Microglia from nGD brains harbored elevated levels of GluCer species (C16, C18, and C20), as well as GlcSph. Notably, the immune cell infiltrates in the nGD brains were also enriched in glucosylceramides (C16 and C18) (*Figure 1—figure supplement 1C*).

For de novo characterization of the brain immune cell microenvironment, sorted CD45+ cells from nGD and control brains were analyzed by scRNA-seq. Dimensionality reduction using t-distributed stochastic neighbor embedding (t-SNE) analysis revealed 15 distinct cellular clusters of CD45+ cells (numbered 0–14) (*Figure 1C*). We assigned these clusters to individual immune subsets based on the expression of known marker genes (*Figure 1C*, *Figure 1—figure supplement 1D and E*). There were major alterations in the immune cell composition in nGD brains, which was qualitatively and quantitatively dominated by various lymphocyte populations (T cells, γδT cells, Treg cells, NK cells, and ILC-2), pDC, and a heterogenous myeloid compartment (cDC, monocytes/macrophages, and granulocytes) (*Figure 1C*, *Figure 1—figure supplement 1D*). Homeostatic microglia were reduced in nGD brains, consistent with impaired microglial homeostasis in nGD pathology (*Figure 1—figure supplement 1A and D*). Interestingly, the B cell population (cluster 4) appeared more pronounced in nGD brains, although it was also present in controls. The microglia and immune cell infiltrates in nGD brains exhibited a striking upregulation of type 1 interferon signature genes (ISG) (*Figure 1D*). Total RNA-seq analysis performed on flow-sorted microglia from nGD and control mice revealed distinct gene expression profiles in GluCer/GlcSph-laden nGD microglia vs. control microglia (*Figure 1—figure supplement 2A*). Microglia of nGD mice exhibited clear downregulation of homeostatic genes and concomitant upregulation of disease-associated microglia (DAM) signature genes, in addition to upregulation of ISGs (*Figure 1—figure supplement 2B and C*). Together, these findings establish GluCer/GlcSph-laden DAM, and peripheral immune cell infiltration as key features of nGD neuropathology.

### *Gba* restoration in microglia and neurons prolongs survival of nGD mice

To dissect the contribution of microglia and neurons in nGD neurodegeneration, we generated nGD *Cx3cr1*^Cre/+ and nGD *Nes*^Cre/+ mice for selective rescue of *Gba* in microglia and neurons, respectively (*Figure 2A*, *Figure 2—figure supplement 1A*). Unless otherwise stated, we used littermate *Gba*^wt/wt with the respective Cre recombinases as controls. Notably, restoration of *Gba* in microglia (nGD *Cx3cr1*^Cre/+ mice) led to more than twofold increase in survival compared to nGD mice (*Figure 2B*). Selective restoration of *Gba* in neurons (nGD *Nes*^Cre/+ mice) further enhanced the survival up to ~200 days where they reached the humane end point (*Figure 2B*). To assess the alterations in microglia and macrophage phenotypes associated with targeting of *Gba* in microglia/neurons, we compared the brains of nGD, nGD *Cx3cr1*^Cre/+, and nGD *Nes*^Cre/+ mice. Restoration of *Gba* in microglia of nGD mice (nGD *Cx3cr1*^Cre/+) resulted in increased homeostatic microglia with reduction in CCR2+ MFs and peripheral immune cell infiltration compared to nGD brains (*Figure 2C*, *Figure 2—figure supplement 1B*). In contrast, restoration of *Gba* in neurons of nGD mice (nGD *Nes*^Cre/+) blocked attrition of microglia as well as the infiltration of CCR2+ MFs and peripheral immune cells (*Figure 2C*). Correspondingly, induction of intracellular Pro-IL-1ß in microglia, an indicator of microglial activation, was observed in both nGD and nGD *Cx3cr1*^Cre/+ brains (*Figure 2D and E*) but not in nGD *Nes*^Cre/+ brains (*Figure 2D and F*, *Figure 2—figure supplement 1C*). Collectively, these data establish the critical role of *Gba* deficiency in neurons in aiding vigorous microglial activation and immune cell infiltration in nGD and nGD *Cx3cr1*^Cre/+ brains. Consistent with this notion, compared to nGD mice, both nGD *Cx3cr1*^Cre/+ mice and

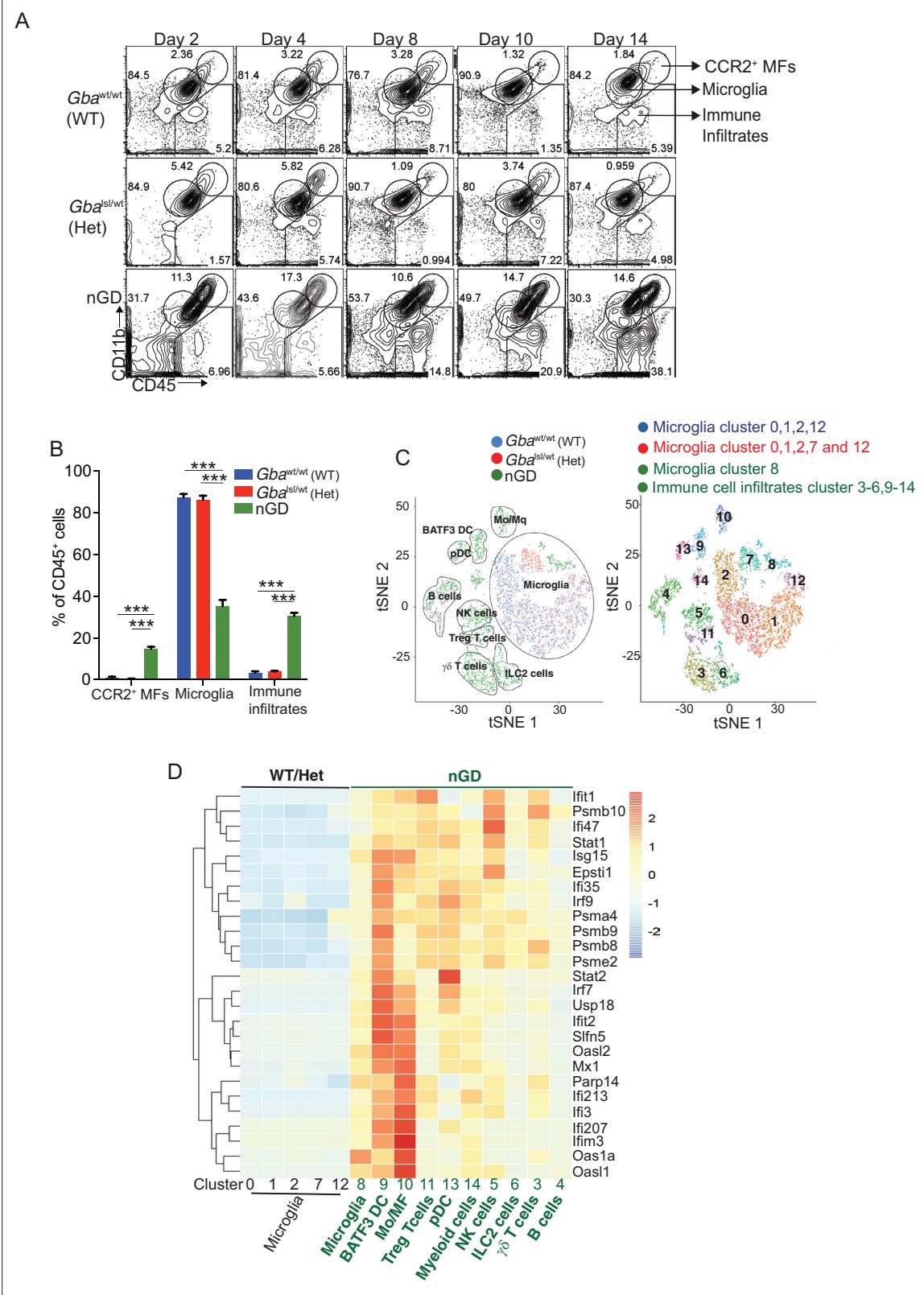

**Figure 1.** Loss of *Gba* induces microglial activation and immune cell infiltration in neuronopathic Gaucher disease (nGD) brain. (**A**) Fluorescence-activated cell sorting (FACS) analysis of the whole brain of *Gba*^wt/wt^, *Gba*^lsl/wt^, and nGD (*Gba*^lsl/lsl^) mice performed at indicated days. The gates indicate cell populations revealed by CD11b and CD45 expression: CCR2+ MFs (CD11b^hi^CD45^+^), microglia (CD11b^lo^CD45^+^), and immune infiltrates (CD11b^lo^CD45^hi^). The data is representative of three independent experiments. (**B**) Bar graph compares percentage of CCR2+ MFS, microglia, and immune infiltrates

Figure 1 continued

between $Gba^{wt/wt}$, $Gba^{lnl/wt}$, and nGD mice brain (n = 6–8 mice/group); statistical significance was determined using *t*-test with using Bonferroni–Dunn correction for multiple comparisons (***p<0.0001). (**C**) t-distributed stochastic neighbor embedding (t-SNE) plot depicting different microglial and non-microglial cell subsets. The clusters are coded based on their mice affiliation (on left). In total, 14 clusters containing 6 microglia clusters and 9 clusters of immune cells (on right). (**D**) Hierarchical clustering of differentially expressed genes associated with type 1 IFN genes from nGD mice vs. the control mice. p<0.05 was considered significant (two-sided *t*-tests). All individual type 1 IFN genes with significant differential expression are listed on right. Error bars represent means ± SEM; p-values were calculated with Welch's test (**p<0.001 and ***p<0.0001).

The online version of this article includes the following figure supplement(s) for figure 1:

**Figure supplement 1.** Loss of *Gba* disrupts microglial homeostasis and induces damage-associated microglia (DAM) phenotype and immune cell infiltration in neuronopathic Gaucher disease (nGD) (*Gba*$^{lsl/lsl}$) brain.

**Figure supplement 2.** Total RNA-seq analysis performed on flow-sorted microglia in neuronopathic Gaucher disease (nGD) (*Gba*$^{lsl/lsl}$) and wild-type mice.

nGD $Nes^{Cre/+}$ mice showed striking reduction of brain glucosylceramides (by ≥90% for C16 and C18:1, and by 50–70% for C18, C20:1, C22, C22.1, C24, and C24:1, and by 38% for C20 glucosylceramide) with a concomitant, striking reduction of GlcSph (89–98%, shown in **Figure 2G**). We depict this data, as fold elevation compared to wild-type mice brains, of various glucosylceramides in nGD and after *Gba* rescue in microglia and neurons in **Figure 2—figure supplement 2A**, which provides insight into relative contributions of microglia and neurons in the accumulation of the lipids in nGD. First, in *Gba* deficiency, the microglia display impressive capacity to process accumulating glucosylceramides, and second, that the major source of accumulating glucosylceramides in nGD appears to be the neuronal compartment. This is illustrated by the observation that compared to wild-type mice brains, in nGD mice, there is a 200-fold elevation of brain C16-glucosylceramide, which falls to 5-fold elevation after microglia rescue and to 2.4-fold after neuronal rescue of *Gba*. Similar gradations are seen for brain GlcSph accumulation: 200-fold vs. 20-fold vs. 7-fold, respectively. Together, these observations suggest that reduced sphingolipid turnover observed in nGD $Nes^{Cre/+}$ mice may exert a positive effect on overall microglial maintenance regardless of microglia cell intrinsic *Gba* deficiency. It seems likely that diverse inflammatory responses triggered by sphingolipid-induced neuronal damage observed in nGD and nGD $Cx3cr1^{Cre/+}$ brains could directly impact microglial maintenance and cell death.

As expected, there was no change in the levels of galactosylceramide (GalCer) species or galactosylsphingosine (GalSph) in nGD brains (**Figure 2—figure supplement 2B**). MALDI imaging of frozen brain sagittal sections was performed to assess the topography of GSL accumulation. Elevated levels of numerous hexosylceramides (HexCers) were detected in nGD brains. Of these, HexCer (18:1/22:0) exhibited higher signal intensity in the cerebral cortex and midbrain region of nGD $Cx3cr1^{Cre/+}$ brains compared to control mice (**Figure 2—figure supplement 2C**). Similarly, HexCer (18:1/20:0) species was elevated in the same brain regions of nGD $Nes^{Cre/+}$ mice (**Figure 2—figure supplement 2D**). Interestingly, a co-regulated lipid in GD cell models, lysophosphatidylcholine (LysoPC) (**Bodennec et al., 2002**), showed striking accumulation in the cerebral cortex and midbrain region of nGD $Cx3cr1^{Cre/+}$ and nGD $Nes^{Cre/+}$ mice (**Figure 2—figure supplement 2F and G**, lower panel). Assessment of motor balance and coordination showed that the longer-lived adult nGD $Nes^{Cre/+}$ (4–6 months) mice were unable to complete the balance beam runs with an average time >60 s (**Figure 2—figure supplement 2E**). Notably, there were unusual repetitive circling body movements (**Figure 2—figure supplement 2F**). These features appear to phenocopy behavioral disorders reported in human GD3 (**Abdelwahab et al., 2017**). Taken together, these results indicate the differential role of *Gba* in individual cell types in the brain. Notably, reconstituting *Gba* in microglia of nGD mice affords an impressive capacity to offset toxic lipid accumulation in the brain and significantly prolong survival.

## Selective deletion of *Gba* in microglia results in late-onset neurodegeneration

To further establish the role of *Gba* in microglia, we used our *Gba* floxed mice (**Mistry et al., 2010**) for microglia-specific *Gba* deletion using *Cx3cr1*-Cre (**Yona et al., 2013**) (*Gba*$^{fl/fl}$*Cx3cr1*$^{Cre/+}$) (**Figure 3—figure supplement 1A**). We investigated this mouse strain for clinical phenotype, brain GluCer/GlcSph accumulation, neuroinflammation, and whether microglia/macrophages underwent compensatory recycling in the setting of microglia-specific *Gba* deficiency. Remarkably, *Gba*$^{fl/fl}$*Cx3cr1*$^{Cre/+}$ brains showed a pronounced accumulation of GlcSph and several GluCer species (C16, C18, C20, and C22 GluCer species) compared

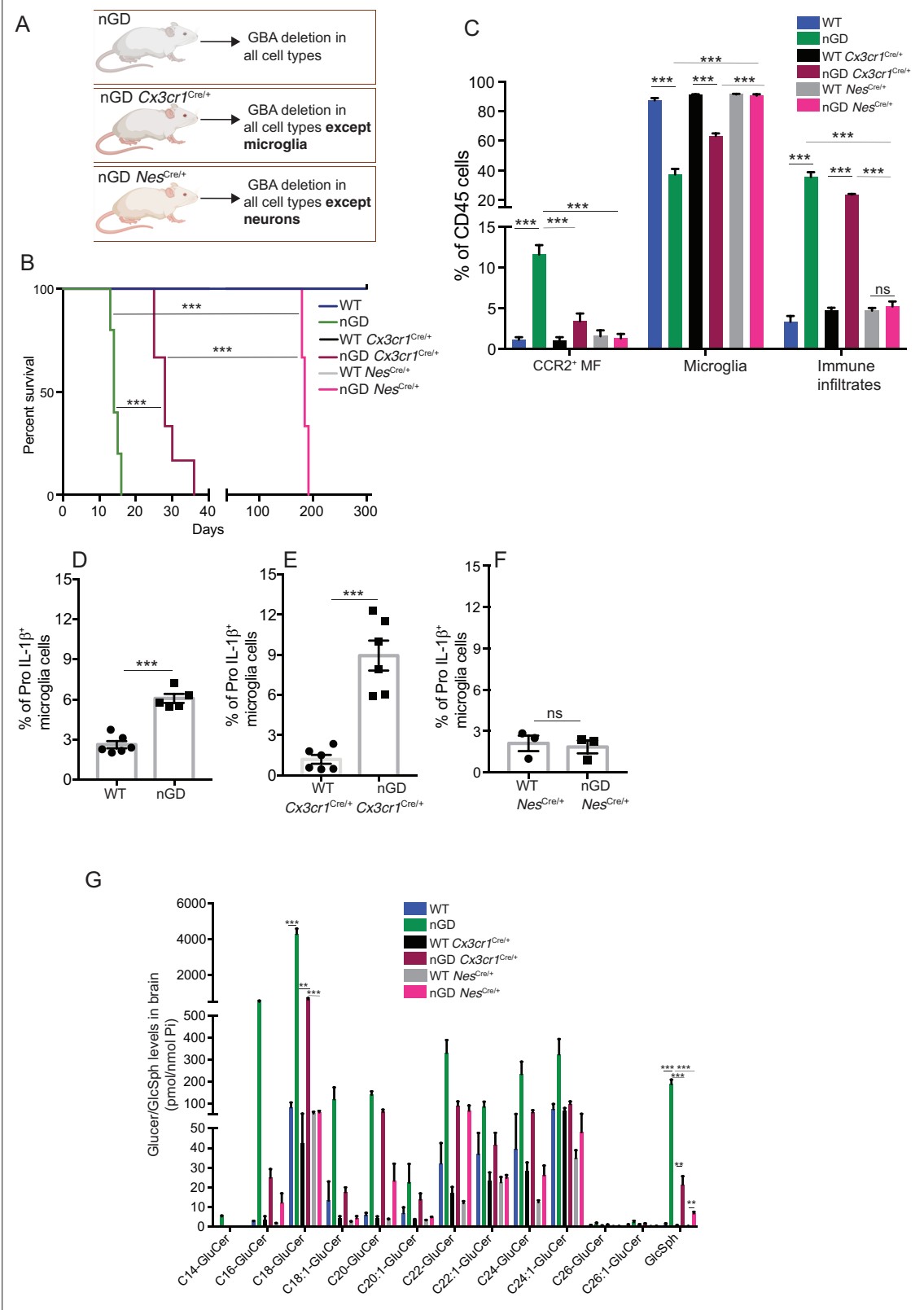

**Figure 2.** *Gba* deficiency in neurons aid in microglial activation and immune cell infiltration. (**A**) Schematic showing overview of the mouse models and methods used in the study. (**B**) Kaplan–Meier survival analysis of neuronopathic Gaucher disease (nGD), nGD *Cx3cr1*^Cre/+, and nGD *Nes*^Cre/+ mice cohorts with their respective littermate controls (n = 5–6 mice/group) using log-rank (Mantel–Cox) test (***p<0.0001). (**C**) Bar graph shows comparison of percentage of CCR2^+ MFs, microglia and immune infiltrates between nGD, nGD *Cx3cr1*^Cre/+, and nGD *Nes*^Cre/+ mice with littermate controls (n =

*Figure 2 continued on next page*

*Figure 2 continued*

3–6 mice/group). Bar graph showing percentage of Pro-IL-1ß⁺ microglia cells in (**D**) nGD vs. control mice (n = 5–6/group), (**E**) nGD *Cx3cr1*^Cre/+ vs. control mice (n = 5–6/group), and (**F**) nGD *Nes*^Cre/+ vs. the control mice (n = 3 mice/group). (**G**) Quantitative analysis of total glucosylceramide (GluCer) species and glucosylsphingosine (GlcSph) levels by LC-ESI-MS/MS in nGD, nGD *Cx3cr1*^Cre/+ mice, and nGD *Nes*^Cre/+ mice brain compared with the control mice (n = 4–8 mice/group). (**C–G**) show representative data from two independent experiments using controls. Means ± SEM are shown. Unpaired *t*-test, two-tailed was used to test significance. *p<0.05, **p<0.001, and ***p<0.0001.

The online version of this article includes the following source data and figure supplement(s) for figure 2:

**Figure supplement 1.** Restoring *Gba* function in microglia and neurons enhances neuronopathic Gaucher disease (nGD) mice survival.

**Figure supplement 1—source data 1.** Gel image showing genotype of neuronopathic Gaucher disease (nGD), Het, and wild-type (wt).

**Figure supplement 1—source data 2.** Gel image showing genotype of neuronopathic Gaucher disease (nGD) *Cx3cr1*^Cre/+ and wild-type (wt) mice.

**Figure supplement 1—source data 3.** Gel image showing genotype of nGD *Nes*^Cre/+ mice and wild-type (wt) mice.

**Figure supplement 1—source data 4.** Gel image showing removal of lsl (lox-stop-lox) cassette in neuronopathic Gaucher disease (nGD) mice in various tissues due to *Krt14*^cre expression.

**Figure supplement 1—source data 5.** Gel image showing Cx3cr1^Cre enabled removal of the lsl (lox-stop-lox) cassette in different tissues of neuronopathic Gaucher disease (nGD) *Cx3cr1*^Cre/+ mice.

**Figure supplement 1—source data 6.** Gel image showing *Nes*^Cre enabled removal of the lsl (lox-stop-lox) cassette in brain of neuronopathic Gaucher disease (nGD) *Nes*^Cre/+ mice.

**Figure supplement 2.** Analysis of lipids in neuronopathic Gaucher disease (nGD), nGD *Cx3cr1*^Cre/+, and nGD *Nes*^Cre/+ mice brain and behavioral studies.

to control brains (*Figure 3A*). There was no change in the levels of GalCer or GalSph species (*Figure 3—figure supplement 1B*). Young mice appeared healthy despite accumulation of GluCer/GlcSph in the brain, but aged *Gba*^fl/fl*Cx3cr1*^Cre/+ mice (~12 months old) started to exhibit motor deficits manifested by a longer time to complete the beam walk and increased tendency to slip (data not shown). HPLC-MS/MS analysis revealed elevated levels of GlcSph in the sera of young *Gba*^fl/fl*Cx3cr1*^Cre/+ mice (6–8 weeks), which tended to increase further in aged (14 months) mice compared to healthy controls (*Figure 3B*). Congruently, there was a significant reduction in microglia and increase in CCR2⁺ MFs and immune cell infiltration in aged *Gba*^fl/fl*Cx3cr1*^Cre/+ mice, while no changes were observed in young *Gba*^fl/fl*Cx3cr1*^Cre/+ mice (*Figure 3C*). We attributed increase of infiltrating CCR2⁺ MFs in aged *Gba*^fl/fl *Cx3cr1*^Cre/+ mice brain compared to young mice to higher turnover rate as seen by BrdU uptake assay (*Figure 3—figure supplement 1C and D*). In contrast to nGD mice with florid neuroinflammation, we could not detect Pro IL-1ß induction in microglia of *Gba*^fl/fl *Cx3cr1*^Cre/+ mice (*Figure 3—figure supplement 1E*). We isolated FACS-sorted microglia from young and aged *Gba*^fl/fl *Cx3cr1*^Cre/+ brains to perform scRNA-seq. A total of 2302 microglial cell transcriptomes were subjected to Louvain clustering, resulting in nine transcriptionally distinct microglial states (*Figure 3D*). The proportion of individual microglial clusters were differentially enriched in young vs. aged mice brain (*Figure 3E*). Taking advantage of the single-cell resolution of our data and published microglia-specific lineage genes (*Chen and Colonna, 2021a*; *Chen et al., 2021b*; *Wang et al., 2020*), we identified homeostatic microglial clusters (0, 2, 3, and 6), DAM clusters (1 and 4), a hitherto undescribed *Gba* locus-associated gene cluster (cluster 5, we refer to it as *Gba* cluster because *Mtx1* and *Thbs3* are contiguous with *the Gba* gene), and ISG cluster (cluster 7) (*Figure 3E and F*). In young *Gba*^fl/fl*Cx3cr1*^Cre/+ mice, homeostatic microglia represented by clusters 0, 2, 3, and 6 were the predominant microglia population (*Figure 3F*). Notably, in aged *Gba*^fl/fl *Cx3cr1*^Cre/+ brains, DAM signature clusters (1 and 4), along with *the Gba* locus-associated gene cluster (cluster 5) and interferon-induced gene cluster (cluster 7), represented the major microglial populations (*Figure 3F*). Therefore, with aging transition of homeostatic microglia (cluster 2) into DAM microglia cluster with higher expression of DAM genes (*Apoe*, *Spp1*, *Lpl*, *Ccl3*, and *Cst7*) and lower levels of homeostasis genes (*P2ry12* and *Tmem119*) was evident (*Figure 3E*). Aged *Gba*^fl/fl *Cx3cr1*^Cre/+ microglia also expressed higher levels of chemokines (*Ccl3*, *Ccl4*, *Ccl2*, etc.), inflammatory molecules (*Tnf*, *Il1a*, and *Il1b*), and key DAM signature genes (*Apoe*, *Spp1*, *Lpl*, *Ccl3*, and *Cst7*) (*Figure 3G*). Notably, CCL2/CCR2 interaction mediates the recruitment of CCR2-bearing leukocytes in the brain in several neuroinflammatory diseases (*Mahad et al., 2006*). In this context, aged *Gba*^fl/fl *Cx3cr1*^Cre/+ mice microglia displaying higher expression of Ccl2 and other inflammatory chemokines may mediate infiltration of peripheral CCR2⁺ MFs and leukocytes contributing to neuroinflammation seen in our late-onset neurodegeneration model, akin to that seen in some GD1 patients with aging (*Belarbi et al., 2020*). Interestingly, expression of *Mtx1* and *Thbs3* genes that are assigned to chromosome 1q21 contiguous with *Gba* was enriched in cluster 5 and prominent in aged*Gba*^fl/fl*Cx3cr1* microglia (we have termed these

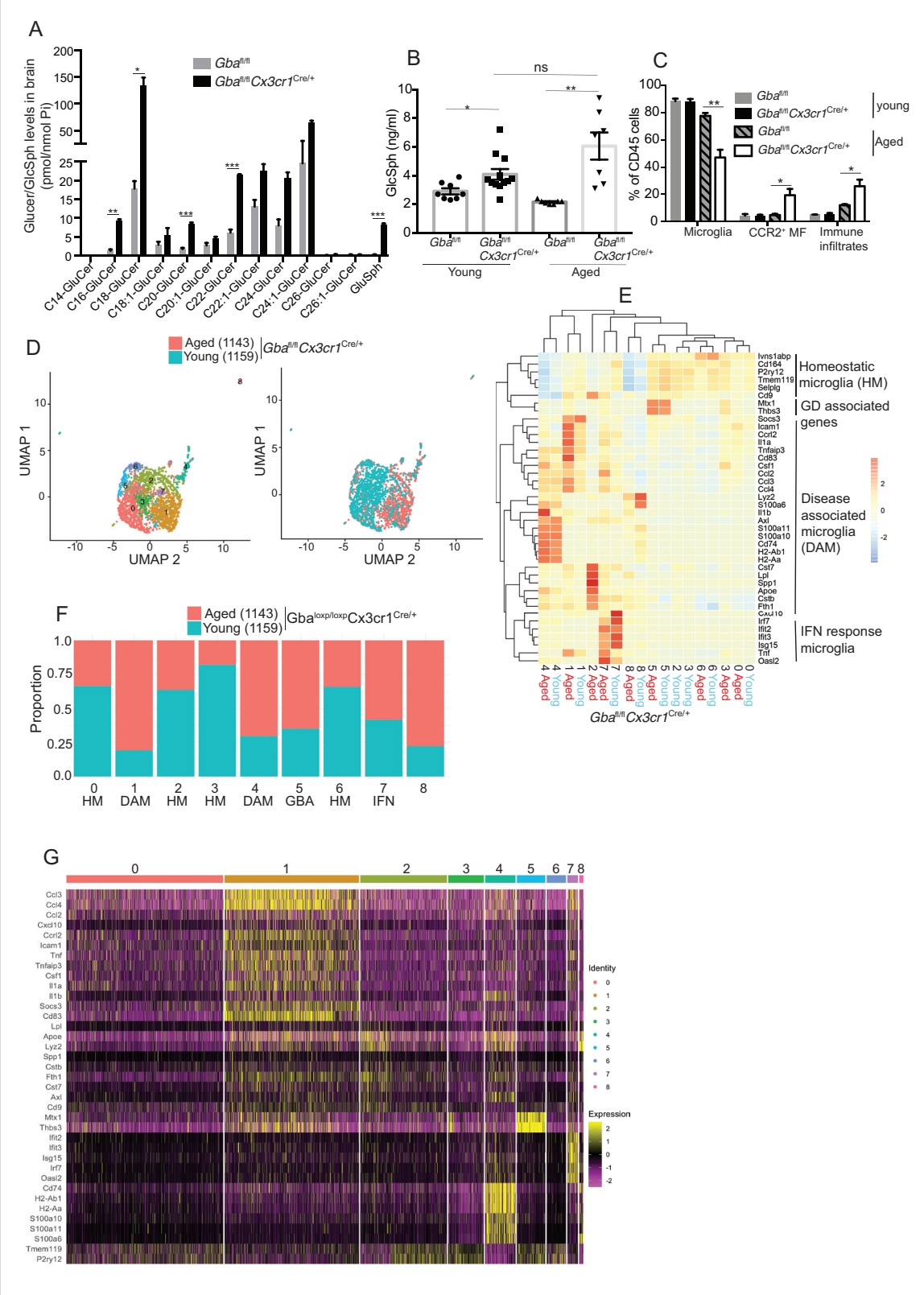

**Figure 3.** Aged $Gba^{fl/fl}Cx3cr1^{Cre/+}$ mice brain show alteration of microglia subsets and neurodegeneration. (**A**) Quantitative analysis of total glucosylceramide/glucosylsphingosine (GluCer/GlcSph) levels in $Gba^{fl/fl}Cx3cr1^{Cre/wt}$ mice and control mice (n = 3 mice/group). Statistical significance was determined using *t*-test using Bonferroni–Dunn correction for multiple comparisons (*p<0.05, **p<0.001, and ***p<0.0001). (**B**) Quantitative analysis of serum GlcSph levels in young and aged $Gba^{fl/fl}Cx3cr1^{Cre/wt}$ mice and control mice (n = 8–11 mice/group; repeated at least three times). (**C**) Bar graph

*Figure 3 continued on next page*

*Figure 3 continued*

shows comparison of percentage of CCR2$^+$ MFs, microglia, and immune infiltrates between young and aged *Gba*$^{fl/fl}$*Cx3cr1*$^{Cre/wt}$ mice and control mice (n = 3 mice/group; repeated at least three times). (**D**) UMAP plots show clustering of microglia from young and aged *Gba*$^{fl/fl}$*Cx3cr1*$^{Cre/wt}$ mice. Cells are colored by cluster (left) and age (right). (**E**) Hierarchical heat map depicting differential expression of genes taken from Wang et al. and compared between microglia cluster from young and aged *Gba*$^{fl/fl}$*Cx3cr1*$^{Cre/wt}$ mice. (**F**) Fraction of cells for each cluster present in young and aged *Gba*$^{fl/fl}$*Cx3cr1*$^{Cre/wt}$ mice, respectively. (**G**) Gene expression heat map for clusters defined as microglia. (**A, B**) Data represents three biological replicates. Means ± SEM are shown. Unpaired *t*-test, two-tailed was used to test significance \*p<0.05, \*\*p<0.001, and \*\*\*p<0.0001.

The online version of this article includes the following source data and figure supplement(s) for figure 3:

**Figure supplement 1.** Changes in microglia subsets coupled with neurodegeneration are seen in aged Gba$^{fl/fl}$ *Cx3cr1*$^{Cre/+}$ mice brain.

**Figure supplement 1—source data 1.** Gel image showing genotype of Gba$^{fl/fl}$ *Cx3cr1*$^{Cre/+}$ mice having conditional deletion of the Gba gene through *Cx3cr1*$^{Cre}$ expression.

---

*Gba*-associated genes). The expression of IFN response genes in microglia cluster 7 remained unchanged between young and aged mice (***Figure 3E***). Taken together, molecular elucidation of aged *Gba*-deficient microglia shows a loss of homeostatic signature with pronounced expression of inflammatory signaling molecules. Thus, our findings highlight the important role of *Gba* in long-term maintenance of microglial homeostasis and implicates *GBA* deficient microglia in the development of age-related neurodegenerative disease like PD that occurs with high risk in adults with GD1.

## NK cells, together with microglial activation, drive neuropathology in nGD brains

Analysis of immune cell infiltrates by scRNA-seq in the brains of nGD and nGD *Cx3cr1*$^{Cre/+}$ mice revealed significant infiltration of NK cells (Fig. S1B and E). We confirmed striking NK infiltration in nGD brains by flow cytometry (***Figure 4A–C***). Brain-infiltrating NK cells expressed granzyme A (GzmA) (***Figure 4D–F***). Interestingly, higher percentages of NK cell infiltration were observed in nGD *Cx3cr1*$^{Cre/+}$ mice brain compared to nGD mice brain. It should be noted that there is increased attrition/death of microglia in nGD mice compared to nGD *Cx3cr1*$^{Cre/+}$ mice. Taking these observations into consideration, we surmise that inflammatory cytokines and chemokines secreted by activated microglia that are present in higher proportion in nGD *Cx3cr1*$^{Cre/+}$ mice induce robust NK cell infiltration into brains than seen in nGD mice. NK cell frequency and GzmA expression were unaltered in the spleens of both nGD and nGD *Cx3cr1*$^{Cre/+}$ mice (***Figure 4—figure supplement 1A***, ***Figure 4G and H***), implying that the primary cues responsible for activation of NK cells were emanating from within the nGD brains. Sphingosine 1-phosphate (Sph-1P) plays a key role in NK cell trafficking via its receptor S1P5 (***Walzer et al., 2007***). To address whether Sph-1P is involved in NK cell infiltration in nGD brain, we performed HPLC-MS/MS analysis of brain tissue for sphingosine species. We found no specific buildup of sphingosine lipid species including Sph-1P in nGD and nGD *Cx3cr1*$^{Cre/+}$ mice brains compared to the control mice (***Figure 4—figure supplement 1B***), suggesting that other mechanism(s) may underlie NK cell infiltration or that there is local generation of S1P that is beyond the resolution of HPLC-MS/MS.

To assess whether the pattern of neuroinflammation observed in genetic models of nGD could be replicated in another system, we used a chemically induced model of nGD using an inhibitor of acid β-glucosidase (CBE). Administration of CBE in WT mice resulted in the nGD phenotype, as described previously (***Farfel-Becker et al., 2011***; ***Kanfer et al., 1982***). FACS analysis of immune cells in the brains of CBE-treated mice revealed major infiltration of immune cells (***Figure 4I***, ***Figure 4—figure supplement 1C***) as well as Pro-IL-1ß induction in the microglia (***Figure 4J***). Like genetic models, immune cell phenotyping in the brains of CBE-treated mice showed striking induction of GzmA$^+$ NK cells (***Figure 4K and L***, ***Figure 4—figure supplement 1C***). Therefore, using genetic and chemically induced nGD mouse models, we confirmed a conserved immune landscape of nGD brains comprising prominent NK cell infiltration and microglial activation.

## Single-cell resolution revealed role of microglial *Gba* in delaying neuroinflammation

To gain a comprehensive and dynamic overview of the underlying mechanisms involved in the longitudinal course of GD-associated neurodegeneration, we performed brain single-nucleus RNA-seq to

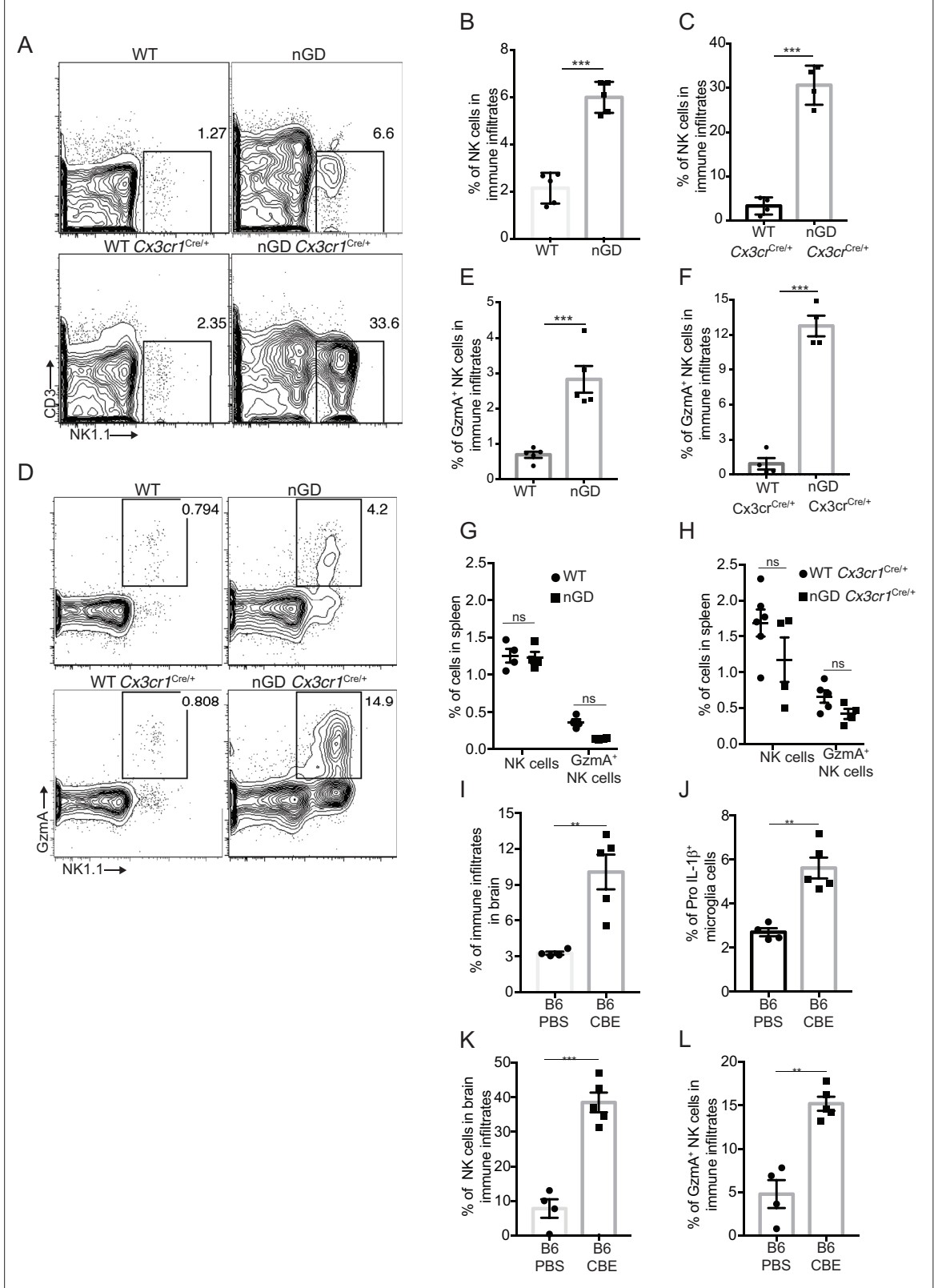

**Figure 4.** NK cell infiltration into the brain of neuronopathic Gaucher disease (nGD), nGD *Cx3cr1*Cre/+ mice. (**A**) CD45+ cells form the whole brain of nGD, nGD *Cx3cr1*Cre/+, and control mice were gated to analyze CD3−NK1.1+ NK cells. Bar graphs represent the percentage of NK cells in the immune infiltrates of (**B**) nGD mice brain (n = 5mice/group) and (**C**) nGD *Cx3cr1*Cre/+ mice brain, respectively (n = 4 mice/group). (**D**) Expression of granzyme A (GzmA) in the brain infiltrating NK cells of nGD and nGD *Cx3cr1*Cre/+ mice. Bar graphs compare the percentage of GzmA+ NK cells in in the immune infiltrates of (**E**)

*Figure 4 continued on next page*

Figure 4 continued

nGD vs. control mice brain (n = 5 mice/group) and (**F**) nGD *Cx3cr1*<sup>Cre/+</sup> vs. control mice brain, respectively (n = 4 mice/group). Percentage of NK1.1[+] NK cells and GzmA[+] NK cells in the spleen of (**G**) nGD mice and (**H**) nGD *Cx3cr1*<sup>Cre/+</sup> mice (n = 4–5 mice/group). Bar graph showing percentage of (**I**) immune infiltrates (**J**) Pro-IL-1ß[+] microglia cells (**K**) percentage of NK cells and (**L**) GzmA[+] NK cells in the whole brain of *Gba*<sup>wt/wt</sup> treated with vehicle or conduritol β-epoxide (CBE). Experiments were repeated thrice. (**A**) and (**D**) were representative from one of the experiments, (**B**), (**C**), (**E–L**) Data are combined from two such experiments. Data are shown as means ± SEM. Unpaired *t*-test, two-tailed was used to test significance. *p<0.05, **p<0.001, and ***p<0.0001.

The online version of this article includes the following figure supplement(s) for figure 4:

**Figure supplement 1.** NK cell infiltration into the brain of neuronopathic Gaucher disease (nGD), nGD Cx3cr1<sup>Cre/+</sup> mice.

elucidate cell types/pathways in three experimental settings: (1) the acute neuropathic model nGD mice at 2 weeks, (2) after restoration of *Gba* in microglia (nGD *Cx3cr1*<sup>Cre/+</sup>) at early asymptomatic stage (2 weeks) and late stage (6 weeks) when it displays overt neurodegenerative phenotype, and (3) late-onset model, nGD *Nes*<sup>Cre/+</sup> brains with rescue of *Gba* in neuronal compartment (age ~7 months). Integrated hierarchical analysis with quality controls (*Figure 5—figure supplement 1A*) revealed 61 distinct cell clusters that were annotated for major cell types based on the differential expression of canonical marker genes (*Figure 5A*, *Figure 5—figure supplement 1B*). We focused on four distinct gene sets: lysosomal biology, ISGs, chemokines, and ApoE. Lipid metabolism genes are known to be upregulated in DAMs. We focused on ApoE because of its established roles in both lipid transport and neurodegeneration highlighted by GD (*Serrano-Pozo et al., 2021*). AUCell analysis was applied to interrogate these gene set enrichments and stratify different cell types: lysosomal biology (*Abca2, Ap1b1, Ap3s1, Ap4s1, Atp6v0a1, Atp6v1h, Cd63, Cd68, Cln3, Ctsb, Ctsd, Ctsl, Ctss, Galc, Galns, Hexa, Hexb, Lamp, Psap, Slc17a5, Sumf1, Gpr37, Gpr37l1, Mtx1,* and *Thbs3*); interferon signaling genes (*Psmb10, Psmb9, Psmb8, Psma4, Psme2, Oas1a, Oasl2, Oasl1, Isg15, Ifi207, Ifi47, Ifit1, Ifi213, Ifitm3, Ifit3, Ifi35, Epsti1, Parp14, Bst2, Irf7, Irf9, Stat1, Stat2, Usp18, Ifit2, Slfn5,* and *Ifih1*); chemokine genes (*Cxcl10, Cxcl12, Ccl5, Cxcl5, Ccl2, Ccl3,* and *Ccl7*); and ApoE. In the brains of nGD and 6-week-old nGD *Cx3cr1*<sup>Cre/+</sup> mice both with overt neurodegenerative phenotypes, lysosomal biology, ISG, and ApoE genes were highly upregulated in microglia, Purkinje, oligodendrocytes (clusters 2, 34, and 36), astrocytes (clusters 22, 23, 4, 48, 52, and 9), and neurons (0, 3, 5, and 6) (*Figure 5B*). In contrast, microglia of 2-week-old nGD *Cx3cr1*<sup>Cre/+</sup> mice were similar to controls. In line with these observations, DAM markers were significantly enriched in microglia from both nGD and 6-week-old nGD *Cx3cr1*<sup>Cre/+</sup> compared to 2-week-old nGD *Cx3cr1*<sup>Cre/+</sup> mice (*Figure 5—figure supplement 1C*), consolidating the importance of microglial *Gba* in ameliorating neurodegeneration seen in florid acute neuroinflammation. Importantly, the results suggest that even in setting of normal microglia *Gba* in nGD *Cx3cr1*<sup>Cre/+</sup> mice, increased flux of glucosylceramide lipids arising from *Gba*-deficient neuronal cells leads to DAM over time as seen at 6 weeks but not at 2 weeks in these mice. *Gba* rescue in the neuronal compartment resulted in the normalization of pathways associated with lysosomal biology genes, ISG genes, chemokine genes, and ApoE in microglia, Purkinje, oligodendrocytes, astrocytes, and neuron cell clusters, and in other brain cell clusters (*Figure 5B*). Nevertheless, lower expression of homeostatic microglia markers was observed in nGD *Nes*<sup>Cre/+</sup> microglia compared to control mice (*Figure 5—figure supplement 1D*).

Gene expression analysis revealed that nGD mice exhibited the largest number of differentially expressed genes (DEGs) (n = 19,274), which were strikingly reduced in 2-week-old nGD *Cx3cr1*<sup>Cre/+</sup> (n = 125) and also in nGD *Nes*<sup>Cre/+</sup> brains, n = 256 (*Figure 5C*). Consistent with the neuroinflammation observed in 6-week-old nGD *Cx3cr1*<sup>Cre/+</sup> mice, elevated DEGs in neuronal clusters were seen (n = 7539) compared to 2-week-old nGD *Cx3cr1*<sup>Cre/+</sup> mice (n = 125) (*Figure 5C*). However, 6-week-old nGD *Cx3cr1*<sup>Cre/+</sup> mice show a reduction of DEGs in different neuronal clusters compared to nGD mice (*Figure 5D*). Collectively, these results indicate the underlying role of microglial *Gba* in safeguarding and delaying neuroinflammatory gene networks in nGD brains.

We found evidence of significant activation of several disease pathways in astrocytes (*Figure 5B*). Astrocytosis has been described in immunohistochemistry of mouse and human neuronopathic brains (*Wong et al., 2004*). We analyzed the astrocyte compartment for the disease-associated astrocyte (DAA) gene signature (*Batiuk et al., 2020*; *Habib et al., 2020*). DAA genes were highly induced in astrocyte clusters from nGD mice and in 6-week-old nGD *Cx3cr1*<sup>Cre/+</sup> mice but not in 2-week-old nGD *Cx3cr1*<sup>Cre/+</sup> mice (*Figure 5E*). IPA was performed on neurons, astrocytes, and microglia. IPA revealed enrichment of necroptosis, interferon, IL8, TREM1, and neuroinflammation signaling pathways

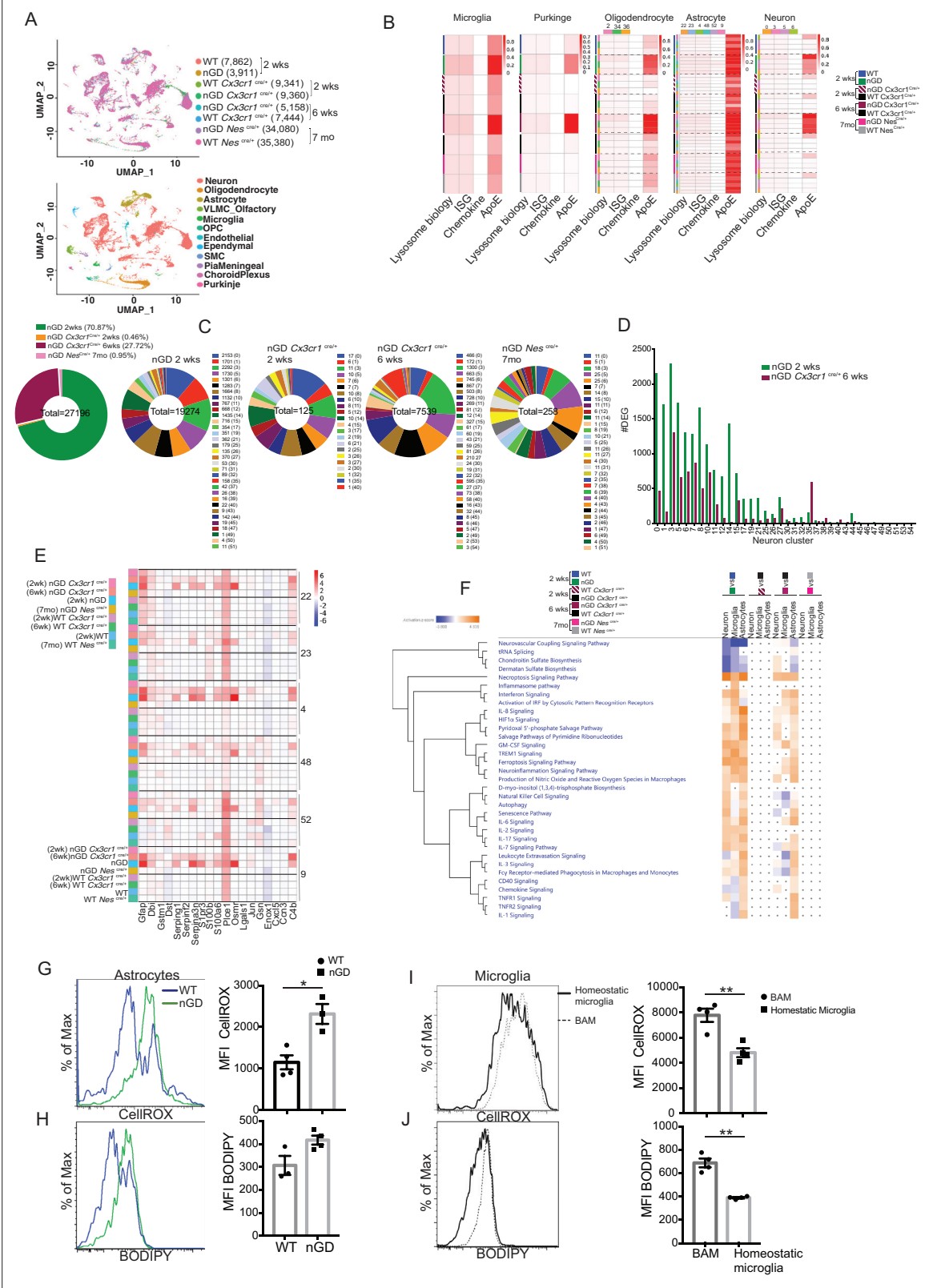

**Figure 5.** snRNA-seq reveals key role of microglia astrocytes and neurons in Gaucher disease (GD)-associated neuroinflammation. (**A**) UMAP data from neuronopathic Gaucher disease (nGD) (2 weeks old), nGD *Cx3cr1*[Cre/+] (2 and 6 weeks old, respectively), nGD *Nes*[Cre/+] mice (n = 3), and corresponding control mice, colored by genotype (top) and cell clusters (bottom). (**B**) AUC analysis for select lysosome, Interferon signature genes (ISG), chemokine, and Apoe gene sets significantly enriched (false discovery rate [FDR] < 0.05) in microglia, Purkinje, oligodendrocyte, astrocyte, and neuro clusters of

*Figure 5 continued on next page*

Figure 5 continued

nGD (2 weeks) vs. control mice; nGD *Cx3cr1*^Cre/+ (2 and 6 weeks, respectively) vs. control mice; nGD *Nes*^Cre/+ mice vs. control mice. Row side bar colors indicate mice genotype, age and cell clusters. (**C**) Pie chart displays the number of differentially expressed genes (DEGs) in neuronal clusters of nGD (2 weeks) vs. control mice (green); nGD *Cx3cr1*^Cre/+ (2 weeks [orange] and 6 weeks [maroon], respectively); nGD *Nes*^Cre/+ mice (pink) vs. control mice with log2(fold change) and adjusted p-value < 0.05. Total DEGs from each set of mice are stated in the middle of pie chart with number of DEGs and the neuronal cluster in brackets shown to the right of each pie chart. (**D**) Bar graph represents the number of DEGs in neuronal clusters of nGD 2 weeks (green) vs. nGD *Cx3cr1*^Cre/+ 6 weeks (maroon) with log2(fold change) and adjusted p-value < 0.05. (**E**) Heat map of DEGs associated with disease-associated astrocytes (DAA) from nGD; nGD *Cx3cr1*^Cre/+ (2 weeks and 6 weeks, respectively); nGD *Nes*^Cre/+ mice vs. control mice. p<0.05 was considered significant (two-sided *t*-tests). All individual DAA genes with significant differential expression are listed on bottom and the astrocyte clusters are shown in right. Red, positive z-score; white, zero z-score; blue, negative z-score. (**F**) Ingenuity pathway analysis (IPA) from nGD vs. WT, nGD *Cx3cr1*^Cre/+ vs. WT *Cx3cr1*^Cre/+ at 2- and 6-week-old mice; nGD *Nes*^Cre/+ mice vs. WT *Nes*^Cre/+ in neuron cluster, microglia, and astrocytes. Orange, positive z-score; white, zero z-score; blue, negative z-score; gray dots are statistically insignificant. (**G**) Representative flow cytometry histogram (left) and quantification of CellROX fluorescence in astrocytes. (**H**) Histogram (left) and quantification (right) of BODIPY fluorescence in astrocytes of nGD mice. (**I**) Flow cytometry histogram (left) and quantification (right) of CellROX fluorescence in activated and homeostatic microglia from nGD mice. (**J**) Histogram (left) and quantification (right) of BODIPY fluorescence in activated and homeostatic microglia from nGD mice n = 3–4 mice per group. Data were replicated in at least two independent experiments. Unpaired *t*-test, two-tailed was used to test significance. *p<0.05, **p<0.001, and ***p<0.0001.

The online version of this article includes the following figure supplement(s) for figure 5:

**Figure supplement 1.** snRNA-seq analysis of the brain of neuronopathic Gaucher disease (nGD), nGD *Cx3cr1*^Cre/+ mice.

in 2-week-old nGD mice compared to the controls. *Gba* rescue in microglia of 2-week-old nGD *Cx3cr1*^Cre/+ mice showed abrogation of neuroinflammatory pathways seen in nGD brains (***Figure 5F***). Between 2 and 6 weeks, as neurodegeneration advanced in nGD *Cx3cr1*^Cre/+ mice, the enrichment of inflammatory pathways was again evident (***Figure 5F***). Notably, both interferon and NK cell signaling pathways were upregulated in 2-week-old nGD and in 6-week-old nGD *Cx3cr1*^Cre/+ brains (***Figure 5F***). Consistent with the lack of microglial activation and NK infiltration in nGD *Nes*^Cre/+ mice, IPA revealed no changes in neurons, astrocytes, and microglia in nGD *Nes*^Cre/+ mice (***Figure 5F***). Pathway analysis of differentially expressed genes revealed that the 'production of nitric oxide and ROS' was elevated in nGD and 6-week-old nGD *Cx3cr1*^Cre/+ microglia and astrocytes along with neuroinflammation pathways. Elevated and dysregulated reactive oxygen species (ROS) production from DAM contributes to oxidative stress and has been shown to be intricately linked with neurodegeneration (***Mendiola et al., 2020***; ***Takizawa et al., 2011***). Consistent with the transcriptional changes observed in nGD mice, astrocytes from nGD mice showed higher fluorescence after treatment with CellROX, indicating a higher level of ROS, compared to astrocytes from control mice (***Figure 5G***). nGD astrocytes with high ROS level also showed elevated accumulation of neutral lipids, as seen by BODIPY staining (***Figure 5H***). Moreover, activated microglia in the nGD brain showed enhanced ROS level along with an increased accumulation of lipids compared to homeostatic microglia (***Figure 5I and J***). Overall, these observations suggest that *Gba*-deficient neurons and microglia, laden with GlcCer and GlcSph, play key roles in orchestrating neuroinflammation involving astrocytes and NK cells that underlie neurodegeneration in neuronopathic GD.

## GCS inhibitor reduces GluCer/GlcSph and reverses microglia and NK cell activation

Collectively our findings suggest that lipid dyshomeostasis and neurodegeneration caused by *Gba* deficiency in microglia and neurons can be prevented by restoration of *Gba*. Previous studies have shown that treatment with GCS inhibitor GZ-161 reduced GluCer and GlcSph in the brains of nGD mice and prolonged survival by a few days (***Cabrera-Salazar et al., 2012***; ***Figure 6—figure supplement 1A***). To determine whether reduction of glucosylceramide lipids by GZ-161 alleviates neuroinflammatory landscape and neurodegeneration, we treated our mouse models with this brain-permeant GCS inhibitor. Consistent with previous studies, GZ-161 treatment significantly prolonged the survival of nGD mice (***Figure 6A***). We investigated whether GZ-161 treatment could further extend the survival of nGD *Cx3cr1*^Cre/+ mice by more effective reduction of brain GluCer/GlcSph via dual effects of decreased synthesis of GluCer and increased lysosomal degradation of GluCer. Indeed, GZ-161 treatment of nGD *Cx3cr1*^Cre/+ mice resulted in considerable extension of survival compared to untreated mice, with analogous normalization of GlcSph levels in the sera of treated nGD *Cx3cr1*^Cre/+ mice (***Figure 6A and B***).

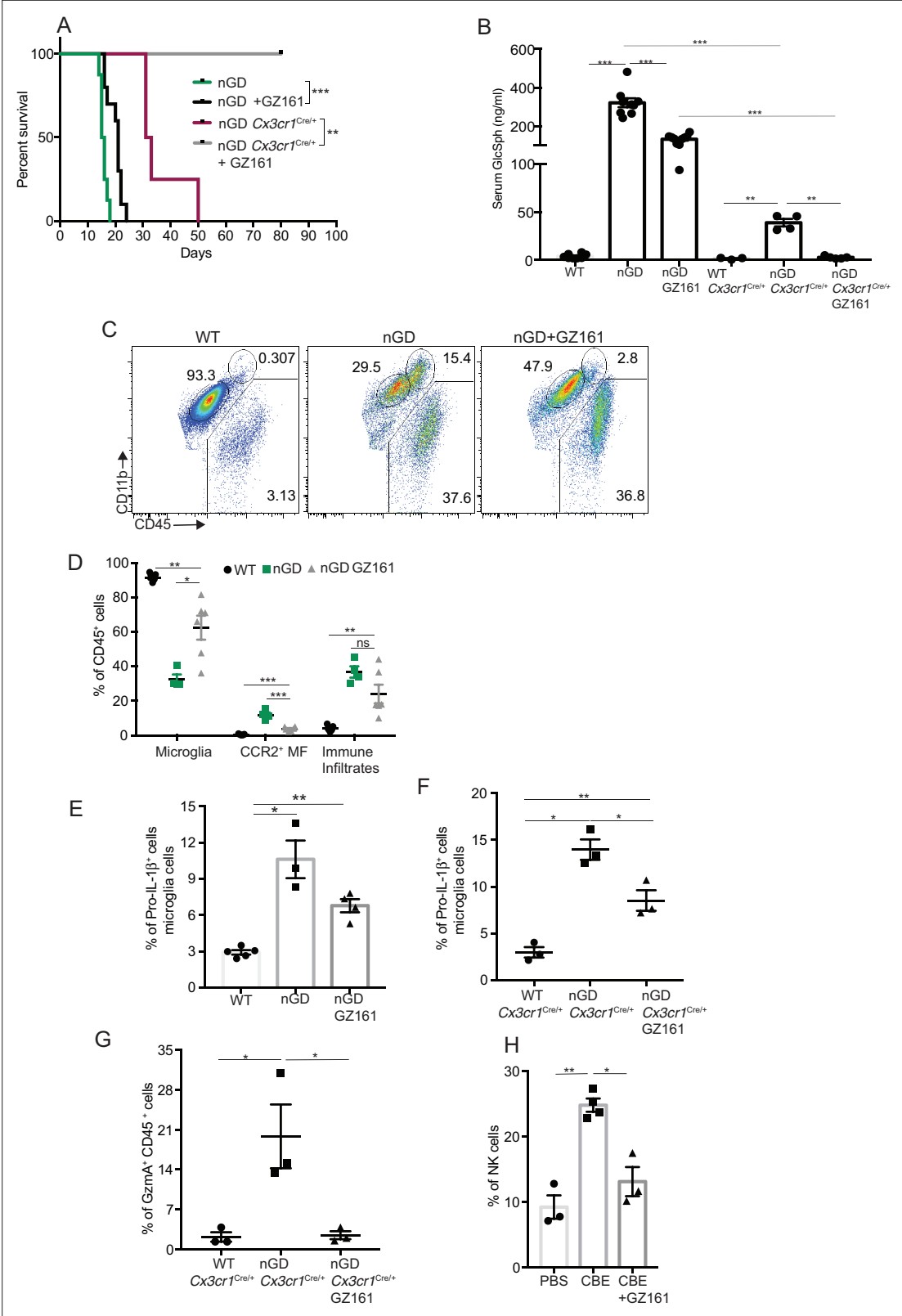

**Figure 6.** Effects of glucosylceramide synthase (GCS) inhibitor (GZ-161) on microglia and GzmA+ cells. (**A**) Kaplan–Meier survival analysis of neuronopathic Gaucher disease (nGD) and nGD *Cx3cr1*Cre/+ mice after GZ-161 treatment compared to vehicle-treated controls (n = 4–10 mice/group). (**B**) Quantitative analysis of serum glucosylsphingosine (GlcSph) levels by LC-ESI-MS/MS in nGD, nGD *Cx3cr1*Cre/+ mice with and without GZ-161 treatment compared with the vehicle-treated controls (n = 4–10 mice/group). (**C**) Representative fluorescence-activated cell sorting (FACS) staining of microglia,

*Figure 6 continued on next page*

*Figure 6 continued*

activated microglia, and immune infiltrates in wild-type and nGD mice with and without treatment with GZ-161. (**D**) Graph represents percentages of microglia, CCR2+ MFs and immune infiltrates in wild-type and nGD mice with and without treatment with GZ-161 (n = 4–6 mice/group). (**E**) Bar graph showing percentage of Pro-IL-1ß+ microglia cells in wild-type and nGD mice with and without treatment with GZ-161 (n = 3–5 mice/group; repeated at least three times). (**F**) Bar graph showing percentage of Pro-IL-1ß+ microglia cells in wild-type and nGD *Cx3cr1*Cre/+ mice with and without treatment with GZ-161 (n = 3–5 mice/group). (**G**) Percentage of GzmA+ CD45+ cells in wild-type and nGD *Cx3cr1*Cre/+ mice with and without treatment with GZ-161 (n = 3–5 mice/group). (**H**) Bar graphs compare the percentage of NK cells in *Gba*wt/wt and mice treated with conduritol β-epoxide (CBE) and CBE + GZ-161, respectively (n = 3–5 mice/group; repeated at least three times). Data represents three biological replicates. Means ± SEM are shown. Unpaired *t*-test, two-tailed was used to test significance. *p<0.05, **p<0.001, and ***p<0.0001.

The online version of this article includes the following figure supplement(s) for figure 6:

**Figure supplement 1.** Immune regulatory effects of glucosyl ceramide synthetase (GCS) inhibitor GZ-161 on microglia and GzmA+ cells.

Next, we assessed the impact of pharmacological reduction of GluCer/GlcSph on disease-specific, microglial and macrophage phenotypes associated with neuroinflammation in GD mice models. FACS analysis of GZ-161-treated nGD mice showed that the GCS inhibitor significantly reduced the proportion of CCR2+ MFs, concomitantly with restoration of homeostatic microglia compartment (*Figure 6C and D*). However, GZ-161 treatment had no effect on immune cell infiltrates in the brains of nGD mice (*Figure 6C and D*). GZ-161 treatment showed a nonsignificant downward trend in microglial Pro-IL-1ß induction in nGD mice (*Figure 6E*). GZ-161 treatment in longer-lived nGD *Cx3cr1*Cre/+ mice wherein drug administration was more reliable resulted in significantly increased survival (*Figure 6A*) accompanied by a reduction in Pro-IL-1ß+ microglia as well as a reduction in GzmA+ immune cells in the brain (*Figure 6—figure supplement 1B*, *Figure 6F and G*). We further evaluated the ability of GZ-161 to counteract NK cell induction seen in the CBE model. Administration of GZ-161 in CBE-treated mice reduced infiltration of NK cells in the brain (*Figure 6H*). Together, these results show that GZ-161 can mitigate the metabolic defect caused by *Gba* deficiency by reducing the accumulation of GluCer and GlcSph, concurrent with alleviating CNS immune inflammation involving microglia and NK cell activation. Moreover, when microglia are predominantly of homeostatic phenotype through complementation of *Gba* function in nGD mice, a two-pronged approach of restoring *Gba* in microglia and GCS inhibitor, the influx of NK cells into the brain and subsequent neurodegeneration is significantly constrained.

## GCS inhibition counteracts age-related microglial dysfunction and NK cell activation

We investigated whether buildup of GlcCer and GlcSph is proximate cause of neuroinflammation and neurodegeneration in *Gba* deficiency using the brain-permeant GCS inhibitor in late-onset neurodegeneration *Gba*fl/fl*Cx3cr1*Cre/+ model. The effect of reversing the toxic accumulation of GluCer/GlcSph on in vivo microglial homeostasis and NK cell activation was assessed by feeding *Gba*fl/fl control mice and *Gba*fl/fl*Cx3cr1*Cre/+ mice with either control or GZ-161-formulated diet, starting at age 3 months, for 7 months. The effect of GZ-161 on GSL accumulation in brain regions was evaluated using MALDI imaging. *Gba*fl/fl*Cx3cr1*Cre/+ mice accumulated HexCer (d18:1/18:0) and HexCer (d18:1/20:0) in the corpus callosum (*Figure 7A*), which was normalized upon GZ-161 diet. Concomitantly, there was a reduction in HexCer (18:1/22:0) accumulation in the cerebral cortex (*Figure 7A*).

We performed scRNA-seq of CD45+ cells from brains of *Gba*fl/fl*Cx3cr1*Cre/+ mice and control mice on control or GZ-161-formulated diet. A total of 40,763 CD45+ single-cell transcriptomes were subjected to unsupervised Louvain clustering resulting in a total of 28 transcriptionally distinct populations comprising 10 broad cell types (*Figure 7B*). Clusters 1, 26, and 16 represented microglia (4230 cells) and were visualized on UMAP, revealing 15 unique clusters (*Figure 7B*, lower panel). We identified clusters 0, 1, and 5 as homeostatic microglia, cluster 2 as DAM, and cluster 7 as *Gba*-associated microglia (*Figure 7C*). DAM cluster 2, which was significantly enriched in *Gba*fl/fl*Cx3cr1*Cre/+ brains, was markedly reduced in *Gba*fl/fl*Cx3cr1*Cre/+ mice fed with GZ-161 diet with concurrent enhancement of homeostatic microglia clusters 0 and 1 (*Figure 7D*). We found that *Gba-associated* gene cluster 7 was present exclusively in *Gba*fl/fl*Cx3cr1*Cre/+ mice both on GZ-161 and on the control diet groups. To assess the effect of GZ-161 on NK cells within the brain environment, we further analyzed cluster 17, which represented NK cells (*Figure 7—figure supplement 1*). GZ-161 treatment resulted in reduction of *Gzma*, *Ccl5*, *Tyrobp*, and *Klf2*, with no change in *Klrd1*, *Klrk1*, *CD52*, and *Ccl4* (*Figure 7E*). Collectively, these results demonstrate that GCS inhibitor-mediated reduction

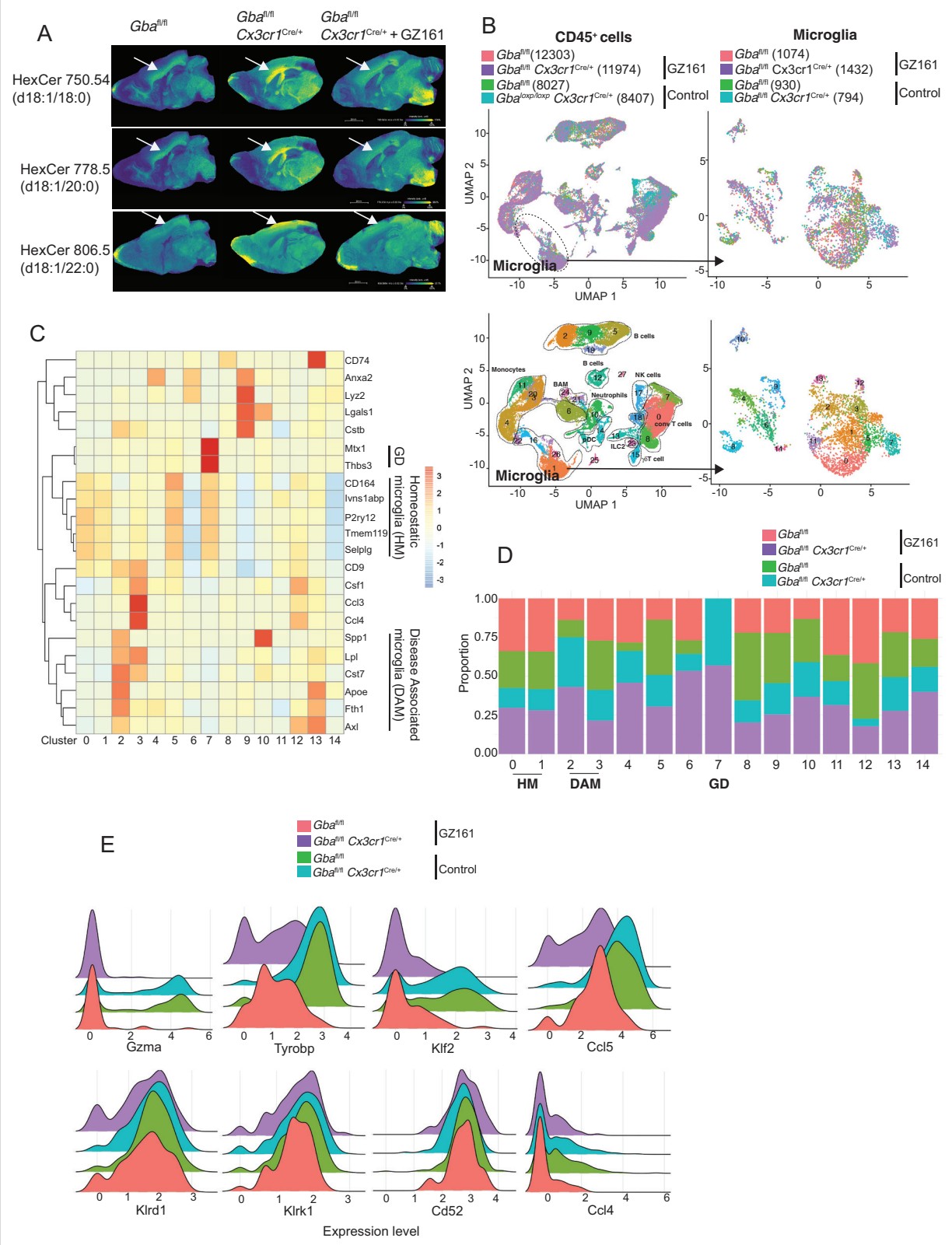

**Figure 7.** Long-term treatment with glucosylceramide synthase (GCS) inhibitor, GZ-161, counteracts age-related microglial dysfunction and NK cell activation. (**A**) Signal intensities of hexosylceramides (HexCer) species and lysophosphatidylcholine (LysoPC) identified by MALDI across *Gba*fl/fl*Cx3cr1*Cre/wt mice treated with either vehicle or GZ161 and control mice brain. The color bars in MALDI images show signal intensity: blue to yellow indicates low to high levels. (**B**) UMAP plots show clusters of CD45+ cells from brain of control and *Gba*fl/fl*Cx3cr1*Cre/wt mice treated with either

*Figure 7 continued on next page*

*Figure 7 continued*

GZ161 or vehicle. Cells are colored by mice genotype (left top) and cluster (left bottom). Microglia subcluster of CD45$^+$ cells, colored by mice (right top) and clusters (right bottom). (**D**) Hierarchical heat map depicting differential expression of genes associated with homeostatic and disease-associated microglia in the different microglia clusters. (**D**) Fraction of cells present in each microglial cluster from control and *Gba*$^{fl/fl}$*Cx3cr1*$^{Cre/wt}$ mice treated with either GZ161 or vehicle. (**E**) Histogram showing differential expression of selected genes in cluster 17 from control and *Gba*$^{fl/fl}$*Cx3cr1*$^{Cre/wt}$ mice treated with either GZ161 or vehicle.

The online version of this article includes the following figure supplement(s) for figure 7:

**Figure supplement 1.** Long-term treatment with glucosylceramide synthase (GCS) inhibitor, GZ-161, counteracts glucosylceramide (GluCer) accumulation, improves microglial homeostasis, and abrogates NK cell activation in *Gba*-deficient microglia.

of GluCer and GlcSph counteracts age-related microglial dysfunction and NK cell activation in several brain regions in *Gba*$^{fl/fl}$*Cx3cr1*$^{Cre/+}$ mice. Together, these findings not only validate GlcCer/GlcSph as the proximate cause of neuroinflammation involving microglia and NK cells but also point to the potential of brain permeant small-molecule inhibitor as a therapeutic approach for neuronopathic GD.

## Generating novel candidate biomarkers of neurodegeneration in *Gba* deficiency

Given the striking relationship between serum Nf-L, level of pathogenic lipid, GlcSph, and neurodegenerative phenotype, we sought to validate these biomarkers in translational studies in GD patients and explore other potential biomarkers suggested by our results. Lack of circulating biomarkers for pre-symptomatic *Gba* deficiency-associated neurodegenerative diseases is a major impediment to optimal management of patients and conduct of clinical trials. Therefore, we investigated whether serum levels of neurofilament light (Nf-L), an accepted biomarker of neuroaxonal injury in several neurodegenerative and neuroinflammatory diseases (*Gaetani et al., 2019*; *Khalil et al., 2020*; *Loeffler et al., 2020*; *Weinhofer et al., 2021*), could serve as a biomarker of neurodegeneration in nGD models. Using ultra-sensitive Quanterix Simoa, we found a massive 2000-fold elevation of serum Nf-L in nGD mice, ~100-fold elevation in nGD *Cx3cr1*$^{Cre/+}$, and ~20-fold elevation in nGD *Nes*$^{Cre/+}$ mice (*Figure 8A*). Remarkably, GZ-161 treatment, which led to reductions in GlcSph levels, also led to a significant decline in serum Nf-L levels in nGD and nGD *Cx3cr1*$^{Cre/+}$ mice (*Figure 8A*). Serum Nf-L level in nGD mice (18,100 ± 4809 pg/ml) was reduced by GZ-161 treatment to 6876 ± 1080 pg/ml. Conversely, GZ-161 treatment of nGD *Cx3cr1*$^{Cre/+}$ mice dramatically reduced Nf-L level to 1297 ± 534.1 pg/ml compared to untreated mice (5466 ± 557.2 pg/ml). The residual Nf-L level after GZ-161 treatment in nGD *Cx3cr1*$^{Cre/+}$ mice was significantly lower than that in nGD mice treated with GZ-161, consistent with the synergistic effect of GCS inhibition and restoration of *Gba* function in microglia (*Figure 8A*). Increased survival of nGD and nGD *Cx3cr1*$^{Cre/+}$ mice after GZ-161 treatment correlated with amelioration of neurodegeneration, as indicated by serum Nf-L levels. Notably, consistent with the age-related progression of neuroaxonal damage, there was a clear age-related increase in serum Nf-L levels with the most significant increase in aged *Gba*$^{fl/fl}$*Cx3cr1*$^{Cre/+}$ mice (*Figure 8B*). Serum levels of Nf-L and GlcSph were strongly correlated ($r$ = 0.8344, p<0.0001), consistent with a direct link between lysosomal generation of toxic GlcSph and neuroaxonal injury (*Figure 8C*). Together, these data support the notion that GlcSph (and GluCer) reduction therapy with the brain-penetrant GCS inhibitor GZ-161 ameliorates neurodegeneration in nGD mice. Moreover, the results suggest that serum Nf-L and GlcSph are promising biomarkers of neurodegeneration in *Gba* deficiency.

Our findings of serum Nf-L correlating with severity of neurodegeneration as well as serum levels of GlcSph in various permutations of nGD mice model for the first time reveal that such circulating biomarkers may exist not only for tracking the severity of neurodegeneration but also for monitoring the response to therapy. Therefore, we conducted an analysis of several candidate biomarkers of neurodegeneration in patients with GD (*Figure 9*). We compared serum Nf-L levels in GD3 patients with early mild neurodegenerative symptoms with age-matched GD1 patients who did not develop early-onset neurodegeneration. Serum Nf-L levels were elevated in GD3 patients compared to those in GD1 (*Figure 8D*). This increase in serum Nf-L levels was associated with elevated serum GlcSph levels in GD3 patients (*Figure 8E*). Akin to age-related increases in serum Nf-L levels in *Gba*$^{fl/fl}$*Cx3cr1*$^{Cre/+}$ mice, indicative of age-related progression of neuroaxonal damage (*Figure 8B*), serum Nf-L levels were significantly elevated in adult GD1 patients compared to children with GD1 and adult healthy controls (*Figure 8F*). This is significant as adult GD1 patients are at an increased risk of PD/

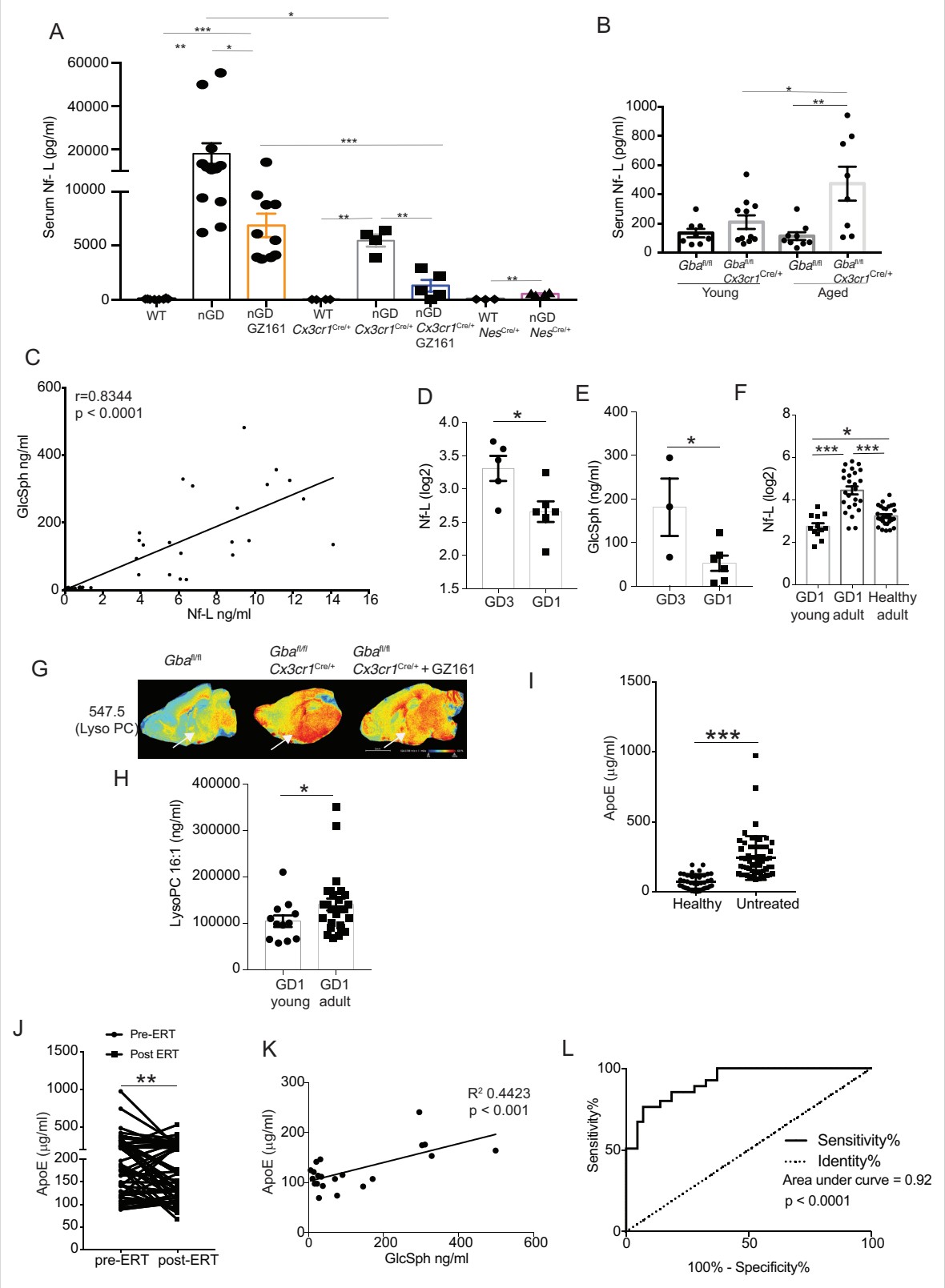

**Figure 8.** Clinical evaluation of ApoE and neurofilament light chain (Nf-L) as biomarkers of Gaucher disease (GD)-associated neurodegeneration. (**A**) Quantitative analysis of serum Nf-L levels in neuronopathic Gaucher disease (nGD), nGD *Cx3cr1*^Cre/+, and nGD *Nes*^Cre/+ mice with and without GZ-161 treatment compared with the vehicle-treated controls (n = 4–10 mice/group; two independent experiments). (**B**) Comparison of serum Nf-L levels between young and aged *Gba*^fl/fl*Cx3cr1*^Cre/wt mice along with control mice (n = 8–11 mice/group). (**C**) Correlation between serum GlcSph and

*Figure 8 continued on next page*

*Figure 8 continued*

Nf-L levels in nGD, nGD *Cx3cr1*<sup>Cre/+</sup> mice, nGD *Nes*<sup>Cre/+</sup>, and control mice. The p-value obtained from Spearman's rank correlation coefficient test was <0.0001 (n = 69 mice). (**D**) Quantitative analysis of serum Nf-L levels (log2 scale) in GD3 patients (n = 5) compared with age-matched GD1 patient (n = 6). (**E**) Quantitative analysis of serum levels glucosylsphingosine (GlcSph) in GD3 patients (n = 3) compared with age-matched GD1 patient (n = 6). (**F**) Quantitative analysis of serum Nf-L levels (log2 scale) in young GD1 patients (n = 12) compared with adult GD1 patient (n = 25) and adult healthy controls (n = 28). (**G**) Signal intensities of lysophosphatidylcholine (LysoPC) identified by MALDI across *Gba*<sup>fl/fl</sup>*Cx3cr1*<sup>Cre/wt</sup> mice treated with either vehicle or GZ161 and control mice brain. The color bars in MALDI images show signal intensity: blue to red indicates low to high levels. (**H**) Quantitative analysis of serum levels of LysoPC 16:1 species in young GD1 patients (n = 12) compared with adult GD1 patient (n = 26). (**I**) Graph represents ApE levels in the sera of untreated GD1 patients (n = 55) and healthy controls (n = 43). (**J**) Graph represents ApoE levels in the sera of untreated GD1 patients (n = 55) and after enzyme replacement therapy (ERT) (n = 55). (**K**) Correlation between serum GlcSph and ApoE levels in sera of GD1 patients. The p-value obtained from Pearson's correlation test was <0.001 (n = 21 patients). (**L**) Receiver-operating characteristic (ROC) curves for serum ApoE expression in GD1 patients and area under the curve (AUC). Means ± SEM are shown. Differences between groups were analyzed using unpaired *t*-test. (**A, B**) Mann–Whitey test (**C, E, H, I**), two-tailed. *p<0.05, **p<0.001, and ***p<0.0001.

The online version of this article includes the following figure supplement(s) for figure 8:

**Figure supplement 1.** Expression of ApoE and *Abca1* in different brain cell types of neuronopathic Gaucher disease (nGD) mice.

LBD. Interestingly, LysoPC was present with an exceptionally strong signal in the microglia in the cerebral cortex and midbrain regions of untreated *Gba*<sup>fl/fl</sup>*Cx3cr1*<sup>Cre/+</sup> mice. Intensity of LysoPC accumulation in brain microglia was significantly reduced by GZ-161 treatment (***Figure 8G***). The finding of striking LysoPC accumulation in *Gba*-deficient microglia prompted us to measure LysoPC in the sera of GD1 patients, comparing young and older patients. Similar to the results with serum Nf-L, older GD1 patients exhibited increased serum levels of LysoPC 16:1 (***Figure 8H***). These findings raise the possibility of broader involvement of lipid metabolism pathways beyond the primary *Gba* defect in GD and fuller understanding could lead to useful biomarkers for the clinic and role of co-regulated lipids in pathophysiology that may be targeted by treatments for GD.

Next, building upon the striking induction of ApoE expression beyond the astrocytes seen in our study, we examined ApoE as a potential biomarker for GD. In steady state, ApoE plays a critical role in lipid metabolism in the brain, with the largest contribution from astrocytes. In disease settings,

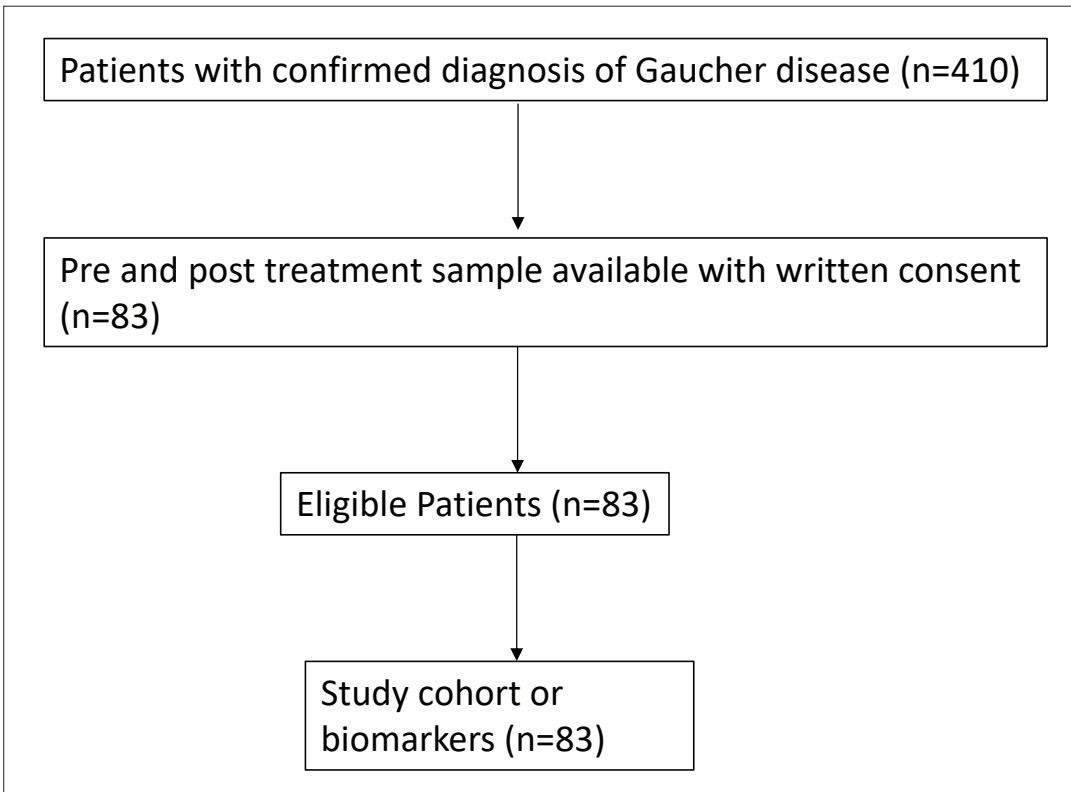

**Figure 9.** Remark flowchart. Flow diagram of the study cohort used in the study.

the induction of ApoE is among the key signature genes of DAM. Thus, the role of ApoE in the brain is multifaceted, including lipid homeostasis, modulation of neuroinflammation, and neuronal repair (*Serrano-Pozo et al., 2021*). Hence, in the context of an inborn error of lipid metabolism exemplified by GD with prominent neurodegeneration, it is highly relevant to explore ApoE expression in various brain cells in different states of nGD. We found prominent ApoE expression in astrocytes and activated microglia in nGD mice. Additionally, there was a striking induction of ApoE in multiple neuronal cell types as well as in endothelial cells (*Figure 8—figure supplement 1A*). Interestingly, there was concomitant induction of *Abca1*, suggesting coupling with increased efflux of accumulating pathogenic lipids from cells in *Gba* deficient brain (*Figure 8—figure supplement 1B*). Therefore, we measured ApoE in the sera of 55 adults with confirmed GD1 before and after ERT, which dramatically lowered GlcSph and reversed systemic clinical manifestations. Sera of untreated GD1 patients showed markedly increased levels of ApoE compared to healthy control sera (*Figure 8I*). ERT resulted in a marked decrease in ApoE levels (*Figure 8J*). Notably, elevated ApoE levels correlated significantly with serum GlcSph levels (*Figure 8K*). The sensitivity and specificity of ApoE as a biomarker were underscored by the area under the receiver operator characteristic (ROC) of 0.92 (p<0.001) (*Figure 8L*), similar to that seen in other accepted biomarkers of GD, such as chitotriosidase, gpNMB, and GlcSph (*Aerts et al., 2011*; *Murugesan et al., 2016*).

## Discussion

There is limited information of how *Gba* deficiency affects the cross-talk between immune cells and neuronal cell types due to accumulating glucosylceramide lipids. Hence, there are no effective therapies for devastating neurodegenerative diseases associated with *Gba* mutations, and clinical trials have been severely hindered by a lack of reliable biomarkers. Previous studies in mice have suggested utility of brain permeant GCS inhibitor as treatment for neuronopathic GD but a clinical trial of miglustat (N-butyldeoxynojirimycin), an iminosugar, did not improve neurological end points in GD3 (*Schiffmann et al., 2008*). Subsequently, more potent GCS inhibitors based on ceramide analogs have shown more promise in genetic nGD model and in chemically induced model (*Blumenreich et al., 2021*; *Cabrera-Salazar et al., 2012*; *Shayman, 2010*; *Wilson et al., 2020*).

To delineate the impact of glycosphingolipid accumulation, identify candidate biomarkers, and assess the impact of GluCer/GlcSph lowering using a potent inhibitor of GCS, we developed an array of genetic mouse models to probe the cell-specific roles of *Gba* at single-cell resolution. Our models recapitulated both early- as well as late-onset neurodegenerative GD that provided unique insights through resolving the heterogeneity of brain macrophage populations. Our findings implicate the essential role of GlcCer-laden microglia and immune infiltrates (including CCR2[+] MFs defined as CD11b[hi] CD45[+]CCR2[+] CD64[+] TIMD4[-] population and NK cells) and astrocytes in orchestrating neuroinflammation. Our studies identified key attributes of GD associated neuroinflammation in the form of attrition of homeostatic microglia, emergence of DAM, influx of CCR2[+] MFs, activation of the ISG pathway, and infiltration of activated NK cells. Massive cellular accumulation of GluCer/GlcSph due to *Gba* deficiency in microglia, immune infiltrates, and neurons resulted in early onset of neuroinflammation, which was attenuated into late-onset neurodegenerative disease by selective rescue of *Gba* in either microglia or neurons as well as by pharmacological reduction of GluCer/GlcSph in the brain using GCS inhibitor. Interestingly, inflammatory microglia also expressed several proposed biomarkers of GD, which we and others have reported such as complement pathway genes, gpNMB, cathepsin D, and cathepsin S (*Afinogenova et al., 2019*; *Kramer et al., 2016*; *Mistry et al., 2014*; *Murugesan et al., 2016*; *Pandey et al., 2017*; *Vitner et al., 2010*; *Zigdon et al., 2015*). Rescue of *Gba* in neuronal progenitors increased survival, but nevertheless, markedly shortened compared to control mice, requiring humane end point sacrifice due to morbid conditions and autistic behavior, reminiscent of human GD3 (*Abdelwahab et al., 2017*; *Bilbo and Stevens, 2017*; *Wong et al., 2004*). Late-onset neurodegenerative disease observed in some patients with GD1 (*Bultron et al., 2010*) appeared to be recapitulated in *Gba*[fl/fl]*Cx3cr1*[Cre/+] mice having slow progressive neuroinflammation with accumulation of GluCer lipids in the brain, elevated serum Nf-L, and DAM gene signatures.

In our nGD mouse models with robust neuroinflammation, we found prominent GzmA[+] NK cell infiltration. Restoring *Gba* function in microglia alone showed limited effect on NK cell infiltration; however, treatment with GCS inhibitor not only offset GluCer/GlcSph buildup, but also exhibited a novel immunomodulatory effect by abrogating NK cell activation. Therefore, combination therapy

with GCS inhibitors and microglia-targeted therapeutics (*Shibuya et al., 2022*) merits further exploration. In general, the immune response of NK cells is fine-tuned by a balance of stimulatory and inhibitory signals based on a distinct receptor repertoire. In certain pathological states, injured neurons display activating NKG2D ligands that target them for NK cell-mediated injury (*Davies et al., 2019*). A potential mechanism relevant to our findings in nGD for NK cell involvement in neurodegeneration is via HLA-1 recognition by NK cells through its inhibitory receptors Ly49 (in mice) and KIR (in humans), which mediate self/non-self-discrimination (*Colonna and Samaridis, 1995*; *Karlhofer et al., 1992*). Thus, downregulation of HLA-1 surface expression is envisioned to trigger NK cell-mediated neuronal injury due to 'missing self.' Consistent with this notion, a recent study reported that altered cell surface GSL repertoire limited accessibility of HLA-1 by immune cells, such as CD8 T cells and NK cells, and treatment with a GCS inhibitor (N-butyl-deoxynojirimycin) fully restored accessibility to HLA-1 (*Jongsma et al., 2021*). Indeed, in our nGD models, a more potent GCS inhibitor, GZ-161, was highly effective in reversing NK cell activation concomitant with marked reduction of brain glucosylceramide lipids. Therefore, it seems likely that in severe *Gba* deficiency, causing early-onset neurodegeneration, altered cell surface GSLs in neurons impair the accessibility of HLA-1 by NK cell receptors, triggering activation and neuronal degeneration.

Our studies show remarkable efficacy of potent brain penetrant GCS inhibitor, GZ-161, in reducing GluCer and GlcSph in immune and neuronal cells of nGD mice concomitant with amelioration of neuroinflammation and neurodegeneration. A prior clinical trial of GCS inhibitor N-butyldeoxynojirimycin in GD3 showed no effect on neurological symptoms (*Schiffmann et al., 2008*). In such clinical trials, patients have established advanced neurological disease that can hinder assessment of full therapeutic effects. Therefore, a major unmet need in neurodegenerative disease associated with *GBA* mutations is the lack of suitable biomarkers to detect neurodegeneration before onset of overt neurological symptoms, such as saccades, ataxia, or seizures. We aimed to leverage the findings from our nGD models to generate novel biomarkers. In our study, neuroaxonal injury was reflected in the elevation of serum Nf-L (*Gaetani et al., 2019*; *Weinhofer et al., 2021*). Data from our nGD models with different rates of neurodegeneration revealed Nf-L as a strong serum biomarker of neurodegeneration that correlated with survival as well as with serum GlcSph. The candidacy of these biomarkers is especially bolstered by the finding of rising levels of serum Nf-L and GlcSph in $Gba^{fl/fl}Cx3cr1^{Cre/+}$ mice as the source of these biomarkers in these mice is from within the brain. These findings are promising to explore serum Nf-L as a biomarker of neurodegeneration to help address significant challenges in the management of patients with GD, that is, distinguish between GD2 and early-onset GD3, track subclinical neurodegeneration in the extremely variable GD3, to help individualize future therapies and identify GD1 individuals at risk for PD/LBD. Indeed, we found that serum Nf-L levels were higher in GD3 patients than in age-matched GD1 patients. Moreover, Nf-L levels were higher in older patients with GD1 compared to younger patients. This is significant considering that adult GD1 patients are at a risk of developing PD/LBD. In this context, our findings of serum Nf-L as a strong biomarker raise the exciting prospect of detecting subclinical neurodegeneration, allowing early initiation of treatment to achieve better clinical outcome. There is an emerging consensus that providing treatment before the onset of overt disease manifestations offers prospects for best outcomes in inborn errors of metabolism. Studying larger cohorts of patients stratified by different types of neurodegenerations due to *GBA* mutations may aid further biomarker validation. The second promising biomarker emanates from striking induction of ApoE in all neuronal cell types beyond astrocytes and DAM in nGD models. Our translational result of striking elevation of ApoE levels in GD patients that correlates with GlcSph levels is compelling to explore the role of ApoE in neurodegeneration associated with *GBA* mutations. Indeed, a recent study showed increased prevalence of Apo E4 allele in heterozygote carriers of GBA mutations who develop PD (*Shiner et al., 2021*).

Overall, through our systematic single-cell transcriptome analysis, we identified cell populations, immunological and pathophysiological mechanisms underlying neurodegeneration associated with *GBA* deficiency. This approach also yielded therapeutic targets and promising biomarkers for clinical validation to improve patient care and aid clinical trials.

## Limitations of the study

There are several limitations to our studies. In $Gba^{lsl/lsl}$ model of fulminant neurodegeneration that phenocopies human GD2, GCS inhibitor prolonged survival significantly but the overall effect was

modest. This is likely related to challenges in administering GCS inhibitor by gavage in the first days of life. While our study showed a critical role for both neuronal and microglial *Gba* in Gaucher-related neurodegeneration and involvement of NK cells, it was beyond the scope of the current work to delineate a hierarchy of neuronal populations according to their vulnerability to toxic accumulation of lipids in setting of *Gba* deficiency and compensatory or pathological interactions with glial cells, infiltrating NK cells, and other immune cells. With regard to application of our findings to human health, an earlier randomized trial of GCS inhibitor, N-butyldeoxynojirimycin was not successful in ameliorating neurological symptoms. However, it is a relatively weak inhibitor of GCS with more potent inhibitory off-target effects (*Schiffmann et al., 2008*). However, more specific and potent GCS inhibitors are showing early promise (*Schiffmann et al., 2020*; *Wilson et al., 2020*).

## Materials availability

All materials in this study are available upon request.

## Acknowledgements

We thank Dr Stefan Karlsson for generously sharing lsl/lsl, nGD mice. We also thank Dr. Sreeganga Chandra for suggestions on mice behavioral tests. This study was funded by a research grant from Sanofi (to PKM).

## Additional information

### Competing interests

Joseph Gans, Erin Teeple: Tri-Hung Nguyen: "ET is an employee of Sanofi. The author has no other competing interests to declare". Sameet Mehta: T-H N was previously an employee of Sanofi. The author has no other competing interests to declare". Martin L Kramer: LG is an employee of Sanofi and holds stocks in Sanofi. The author has no other competing interests to declare. Jiapeng Ruan: MK was previously an employee of Sanofi and holds stocks in Sanofi. The author has no other competing interests to declare. Matthew Davison: HW was previously an employee of Sanofi. The author has no other competing interests to declare. Dinesh Kumar: HW (now retired) was previously an employee of Sanofi and holds stocks in Sanofi. The author has no other competing interests to declare. Bailin Zhang: BZ is an employee of Sanofi. The author has no other competing interests to declare. Katherine Klinger: KK is an employee of Sanofi. The author has no other competing interests to declare. The other authors declare that no competing interests exist.

### Funding

| Funder | Grant reference number | Author |
| --- | --- | --- |
| Sanofi | Research grant from Sanofi | Pramod K Mistry |
| National Institute of Neurological Disorders and Stroke | R01NS110354 | Pramod K Mistry |

The funders had no role in study design, data collection and interpretation, or the decision to submit the work for publication.

### Author contributions

Chandra Sekhar Boddupalli, Conceptualization, Data curation, Formal analysis, Validation, Investigation, Methodology, Writing – original draft, Writing – review and editing; Shiny Nair, Data curation, Formal analysis, Validation, Investigation, Methodology, Writing – original draft, Writing – review and editing; Glenn Belinsky, Jiapeng Ruan, Formal analysis, Methodology; Joseph Gans, Resources, Data curation, Investigation, Methodology; Erin Teeple, Resources, Data curation, Software, Formal analysis; Tri-Hung Nguyen, Lilu Guo, Martin L Kramer, Formal analysis, Investigation; Sameet Mehta, Honggge Wang, Formal analysis; Matthew Davison, Bailin Zhang, Katherine Klinger, Supervision; Dinesh Kumar, Resources, Software, Formal analysis; DJ Vidyadhara,

Methodology; Pramod K Mistry, Conceptualization, Funding acquisition, Project administration, Writing – review and editing

### Author ORCIDs
Chandra Sekhar Boddupalli http://orcid.org/0000-0002-1406-7973
Shiny Nair http://orcid.org/0000-0002-4081-7811
Pramod K Mistry http://orcid.org/0000-0003-3447-6421

### Ethics
Human subjects: Informed consent, and consent to publish, was obtained from the patients and the study was approved by the Human Investigations Committee of Yale School of Medicine.
Mice were housed in the animal facility of Yale university in New Haven. All animal experiments were conducted in compliance with institutional regulations under authorized protocol (2016-10872) approved by the Institutional Animal Care and Use Committee.

### Decision letter and Author response
Decision letter https://doi.org/10.7554/eLife.79830.sa1
Author response https://doi.org/10.7554/eLife.79830.sa2

## Additional files

### Supplementary files
• MDAR checklist

### Data availability
All data generated or analysed during this study are included in the manuscript and supporting file; source data files have been provided for Figure 2—figure supplement 1 and Figure 3—figure supplement 1. Raw data files for Single cell and single nucleus RNA sequencing are deposited with NCBI GEO under accession numbers GSE207765, GSE207766 and GSE207768.

The following datasets were generated:

| Author(s) | Year | Dataset title | Dataset URL | Database and Identifier |
|---|---|---|---|---|
| Boddupalli CS, Mehta S | 2022 | scRNA seq analysis on immune cells isolated from nGD mice brain | https://www.ncbi.nlm.nih.gov/geo/query/acc.cgi?acc=GSE207766 | NCBI Gene Expression Omnibus, GSE207766 |
| Boddupalli CS, Nair S, Belinsky G, Gans J, Teeple E, Nguyen T-H, Mehta S, Guo L, Kramer ML, Ruan J, Wang H, Davison M, Kumar D, Bailin DJZ, Klinger K, Mistry PK | 2022 | RNA and scRNA seq analyses on immune cells isolated from nGD mice brain | https://www.ncbi.nlm.nih.gov/geo/query/acc.cgi?acc=GSE207768 | NCBI Gene Expression Omnibus, GSE207768 |
| Boddupalli CS, Nair S, Belinsky G, Gans J, Teeple E, Nguyen T-HN, Mehta S, Guo L, Kramer ML, Ruan J, Wang H, Davison M, Kumar D, Vidyadhara DJ, Bailin DJZ, Klinger K, Mistry PK | 2022 | RNA seq analysis on immune cells isolated from nGD mice brain | https://www.ncbi.nlm.nih.gov/geo/query/acc.cgi?acc=GSE207765 | NCBI Gene Expression Omnibus, GSE207765 |

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
