## [Editor Report]

The pathophysiology of neuropathic Gaucher disease (nGD) is far from clear. This study not only provides mechanistic insights into the processes underpinning microglial activation and neuroinflammation but also implicates glycosylceramide in the pathogenesis of nGD through the use of a novel inhibitor. Overall, the landmark study substantially improves our understanding of nGD with significant therapeutic ramifications.

---

## [Decision Letter]

**Decision letter after peer review:**

Thank you for submitting your article "Neuroinflammation in neuronopathic Gaucher disease: Role of microglia and NK cells" for consideration by *eLife*. Your article has been reviewed by 2 peer reviewers, and the evaluation has been overseen by Mone Zaidi. The reviewers have opted to remain anonymous.

Essential revisions:

1) Line 142: Define MFs abbreviation.

2) Line 157: Sentence not clear, a connecting word seems to be missing.

3) Legend to Figure 1 lists a panel (H) that is not shown in the Figure.

4) Line 231: In the sentence….. we could not detect Pro IL-1ß induction in microglia of Gbaloxp/loxp Cx3cr1Cre/+ mice (Figure S5E)., the text refers to Figure S5D, not S5E. Correction is needed.

5) There is no further reference to Figure S5E in the text.

6) Revise the legend to Figure S5D and S5E to reflect what is shown in the figure.

7) Line 186. I suggest the authors comment on why restoration of Gba in neurons of nGD mice blocked attrition of microglia (Figure 2C).

8) Figure 4. Is there an explanation for why there is more infiltration of NK cells in nGD than in nGD Cx3cr1Cre/+ mice?

9) Figure 4I. The regimen of CBE administration should be included in the M and M.

10) Line 344/Figure 5F. Do the authors imply that Gba rescue in neurons prevents microglia activation? If so, they should elaborate on that.

11) Line 361. Remove "restoring" from the sentence.

12) Lines 368-371. The authors state that GZ161 increases the survival of nGD Cx3cr1Cre/+ mice by reducing the generation of GSLs in microglia. However, GZ161 acts not only on microglia but on other cells of the brain, including neurons. Therefore this sentence should be modified, as the beneficial effects of the GCS inhibitor may be due to its effects on more than one type of cell present in brain.

13) Figure 6D. Do the grey triangles represent nGD + GZ161? The Figure shows Gbalnl/lnl + GZ161. nGD + GZ161 should be used instead of Gbalnl/lnl + GZ161 for consistency and to avoid confusion.

14) Line 404. …. "Intensity of LysoPC accumulation in brain microglia was significantly reduced by GZ-161 treatment". Are these results not shown?

15) Figure 8A. The authors should comment on the low levels of Nf-L in sera from mice with gba-rescued neurons, as the presence of gba in neurons seems to offset the deleterious effects of its absence in microglia and other tissues. Was GZ161 also tested in nGD NesCre/+ mice?

*Reviewer #1 (Recommendations for the authors):*

This study by Boddupalli et al., was well designed and conducted, the results were reasonably interpreted, and the manuscript was clearly written with logical inputs.

It would further gain the significance of this study by performing brain region or tissue-specific manipulations of the Gba signaling at single-cell resolution, instead of constitutive manipulations.

*Reviewer #2 (Recommendations for the authors):*

In this manuscript, Boddupalli et al., did a careful analysis of the metabolic and molecular alterations caused by Gba deficiency in microglia, neurons, astrocytes, NK cells, macrophages, and other immune cells infiltrating the affected brain. The authors used a number of relevant animal models of neuronopathic Gaucher disease (nGD) that included Gba-null mice with restored Gba expression in either microglia or neurons, and mice in which Gba was deleted only in microglia. These models recapitulated early as well as late-onset nGD. Using single-cell analysis to define alterations in the different cell types present in the brains of Gba mutant mice, the authors clearly demonstrated deregulation of important neuroinflammatory networks at a single cell resolution level. Targeted rescue of Gba in microglia and in neurons of nGD mice reversed the buildup of GSLs with reversal of neuroinflammation and improved survival. The authors further showed that microglia activation was central to neuroinflammation in nGD. The work also identified early biomarkers to follow disease progression and response to treatment, which were validated using sera from patients with neuronopathic GD3 mutations, and in GD1 patients harboring mild mutations. The new markers identified included Nf-L and ApoE, whose levels in serum correlated with those of GluSph, a widely accepted marker of GD. Nf-L was elevated 2,000-fold in nGD mice, and there were also substantial elevations in adult GD1 patients compared to control adults. The authors also found significant differences in these disease markers between young and older GD1 patients. Nf-L and ApoE levels in the mutant mice and nGD patients were significantly reduced by brain-penetrant GluCer synthase inhibitors, further implicating elevated GSLs in the pathogenesis of Gba-associated neurodegeneration, and validating substrate reduction as a useful therapy. Another potential biomarker uncovered in these studies is LysoPC 16:1.

These results of this ground-breaking study are critically important for therapeutic development, as there is an unmet need to identify GD patients and GD carriers at risk for PD/LBD. In sum, the results presented are of great clinical significance, and I would recommend publication without delay.

---

## [Author Response]

Essential revisions:1) Line 142: Define MFs abbreviation.

The abbreviation has been defined in the revised manuscript. See line 384:

CCR2^+^ macrophages (MFs) defined as CD11b^hi^ CD45^+^CCR2^+^ CD64^+^ TIMD4^-^ population

2) Line 157: Sentence not clear, a connecting word seems to be missing.

The sentence has been modified please see line 398 in revised manuscript: For de novo characterization of the brain immune cell microenvironment, sorted CD45^+^ cells from nGD and control brains were analyzed by scRNA-seq. Dimensionality reduction using t-distributed stochastic neighbor embedding (t-SNE) analysis revealed 15 distinct cellular clusters of CD45^+^ cells (numbered 0-14) (Figure 1C). We assigned these clusters to individual immune subsets based on the expression of known marker genes (Figure 1C and figure supplement 1D and E).

3) Legend to Figure 1 lists a panel (H) that is not shown in the Figure.

Legend has been corrected and legend for panel H has been removed. Please see Line # 882.

4) Line 231: In the sentence….. we could not detect Pro IL-1ß induction in microglia of Gbaloxp/loxp Cx3cr1Cre/+ mice (Figure S5E)., the text refers to Figure S5D, not S5E. Correction is needed.

Figure panel as well as Figure3—figure supplement 1 legend has been changed to match the corresponding text. Please see line # 1078-1081 in the revised manuscript.

5) There is no further reference to Figure S5E in the text.

Figure panel as well as Figure3—figure supplement 1 legend has been changed to match the corresponding text. Please see line # 1078-1081 in the revised manuscript.

6) Revise the legend to Figure S5D and S5E to reflect what is shown in the figure.

Figure panel as well as Figure3—figure supplement 1 legend has been changed to match the corresponding text. Please see line # 1078-1081 in the revised manuscript.

7) Line 186. I suggest the authors comment on why restoration of Gba in neurons of nGD mice blocked attrition of microglia (Figure 2C).

As pointed by the reviewer, restoration of Gba in neuronal compartment has significant effect on brain microglial maintenance. Please see lines 441-455 in revised manuscript addressing the reviewers comment:

“We depict this data, as fold elevation compared to wild type mice brains, of various glucosylceramides in nGD and after *Gba* rescue in microglia and neurons in Figure 2—figure supplement 2A which provides insight on relative contributions of microglia and neurons in the accumulation of the lipids in nGD. First, in *Gba* deficiency, the microglia display impressive capacity to process accumulating glucosylceramides and second that the major source of accumulating glucosylceramides in nGD appears to be the neuronal compartment. This is illustrated, by the observation that compared to wild type mice brains, in nGD mice, there is 200-fold elevation of brain C16-glucosylceramide, which falls to 5-fold elevation after microglia rescue and to 2.4-fold after neuronal rescue of *Gba*. Similar gradations are seen for brain GlcSph accumulation: 200-fold vs 20-fold vs 7-fold, respectively. Together, these observations suggest that reduced sphingolipid turnover observed in nGD *Nes*^Cre/+^ mice may exert a positive effect on overall microglial maintenance regardless of microglia cell intrinsic *Gba* deficiency. It seems likely, that diverse inflammatory responses triggered by sphingolipid induced neuronal damage observed in nGD and nGD *Cx3cr1*^Cre/+^ brains could directly impact microglial maintenance and cell death.”

8) Figure 4. Is there an explanation for why there is more infiltration of NK cells in nGD than in nGD Cx3cr1Cre/+ mice?

Higher percentages of NK cell infiltration were observed in nGD *Cx3cr1*^Cre/+^ mice brain as compared to nGD mice brain. It should be noted that there is increased attrition/death of microglia in nGD mice as compared to nGD *Cx3cr1*^Cre/+^ mice. Taking these observations into consideration, we surmise that inflammatory cytokines and chemokines secreted by activated microglia which are present in higher proportion in nGD *Cx3cr1*^Cre/+^ mice induce robust NK cell infiltration into brains than seen in nGD mice. See lines 530-535.

9) Figure 4I. The regimen of CBE administration should be included in the M and M.

Regimen for CBE and GZ161 administration has been included in the M and M. Please see lines 154-159.

10) Line 344/Figure 5F. Do the authors imply that Gba rescue in neurons prevents microglia activation? If so, they should elaborate on that.

We thank the Reviewer for this comment. We have elaborated on this comment in our response #7 above.

Microglial activation was much less in nGD *Nes*
^Cre/+^ mice compared to nGD *Cx3cr1*
^Cre/+^ and nGD, we attribute this higher microglial activation phenomenon to abberant brain sphingolipid turnover seen in both nGD *Cx3cr1*
^Cre/+^ and nGD mice as compared to nGD *Nes*
^Cre/+^ mice.

11) Line 361. Remove "restoring" from the sentence.

The sentence has been corrected.

12) Lines 368-371. The authors state that GZ161 increases the survival of nGD Cx3cr1Cre/+ mice by reducing the generation of GSLs in microglia. However, GZ161 acts not only on microglia but on other cells of the brain, including neurons. Therefore this sentence should be modified, as the beneficial effects of the GCS inhibitor may be due to its effects on more than one type of cell present in brain.

The sentence has been modified according to the reviewer’s suggestion. Please see lines 638-642.

13) Figure 6D. Do the grey triangles represent nGD + GZ161? The Figure shows Gbalnl/lnl + GZ161. nGD + GZ161 should be used instead of Gbalnl/lnl + GZ161 for consistency and to avoid confusion.

Grey triangles do represent nGD + GZ161 and Figure 6D has been modified to reflect the same.

14) Line 404. …. "Intensity of LysoPC accumulation in brain microglia was significantly reduced by GZ-161 treatment". Are these results not shown?

Yes, the data is shown in Figure 8G. The sentence has also been moved to line 731 in the revised manuscript.

15) Figure 8A. The authors should comment on the low levels of Nf-L in sera from mice with gba-rescued neurons, as the presence of gba in neurons seems to offset the deleterious effects of its absence in microglia and other tissues. Was GZ161 also tested in nGD NesCre/+ mice?

As reviewers rightly pointed out the selective knockout of *gba* from neurons using a nestin-cre system does not cause overt microglial attrition or inflammation, nevertheless as shown in (Figure 5—figure supplement 1D), nGD Nes ^Cre/+^ mice show decreased expression of homeostatic microglia markers as compared to littermate wild type control. It is therefore likely that low-grade microglial dyshomeostasis can lead to increase in low levels of Nf-L in sera of nGD Nes ^Cre/+^ mice. In addition to this we believe that absence of Gba function in multiple cell types of the brain could have caused low levels of Nf-L induction in nGD Nes ^Cre/+^ mice.

Regarding reviewers question whether we tested GZ161 in nGD Nes ^Cre/+^ mice, we did not test GZ161 in nGD NesCre/+ mice. We thank the Reviewer for this comment and plan to do this in future experiments.